


# Mid-level convection in a warm conveyor belt accelerates the jet stream

Nicolas Blanchard[1], Florian Pantillon[1], Jean-Pierre Chaboureau[1], and Julien Delanoë[2]

[1]Laboratoire d'Aérologie, Université de Toulouse, CNRS, UPS, Toulouse, France
[2]LATMOS/IPSL, UVSQ Université Paris-Saclay, Sorbonne Université, CNRS, Guyancourt, France

**Correspondence:** Florian Pantillon (florian.pantillon@aero.obs-mip.fr)

**Abstract.** Jet streams and potential vorticity (PV) gradients along upper-level ridges and troughs form a waveguide that governs midlatitude dynamics. Warm conveyor belt (WCB) outflows often inject low-PV air into ridges and their representation is seen as a source of uncertainty for downstream forecasts. Recent studies have highlighted the presence of mesoscale structures of negative PV in WCBs, the impact of which on large-scale dynamics is still debated. Here, fine-scale observations of cloud

5 and wind structures acquired with airborne Doppler radar and dropsondes provide rare information on the WCB outflow of the Stalactite cyclone and the associated upper-level ridge on 2 October 2016 during the North Atlantic Waveguide and Downstream Impact Experiment. The observations reveal a complex tropopause structure with a high PV tongue separating the northwestern edge of the ridge in two parts, each with cirrus-type clouds and accompanied by a jet stream core, and bounded by a tropopause fold. A reference, convection-permitting simulation with full physics reproduces well the observed mesoscale

10 structures and reveals the presence of elongated negative PV bands along the eastern jet stream core. In contrast, a sensitivity experiment with heat exchanges due to cloud processes cut off shows lower cloud tops, weaker jet stream cores, a ridge less extended westward, and the absence of negative PV bands. A Lagrangian analysis based on online trajectories shows that the anticyclonic branch of the WCB outflow feeds the eastern jet stream core in the reference simulation, while it is absent in the sensitivity experiment. The anticyclonic ascents and negative PV bands originate from the same region near the cyclone's

15 bent-back front. The most rapid ascents coincide with mid-level convective cells identified by clustering analysis, which are located in a region of conditional instability below the jet stream core and above a low-level jet. Horizontal PV dipoles are found around these cells and with the negative poles reaching absolute negative values, thus appear as the source of negative PV bands. The results show that mid-level convection within WCBs accelerates the jet stream and may thus influence the downstream large-scale circulation.





# 1   Introduction

Jet streams and potential vorticity (PV) gradients along upper-level ridges and troughs form a waveguide that governs the propagation of Rossby waves (Hoskins and Ambrizzi, 1993). Rossby waves are the main drivers of midlatitude dynamics, constrain the formation of surface cyclones and anticyclones and act as precursors to high-impact weather events. An accurate representation of jet streams and PV gradients is therefore crucial in numerical weather prediction systems. However, it has been found that the PV gradient across the tropopause adjacent to ridges and the amplitude of Rossby waves decrease with lead time in global model forecasts until about 5 days (Gray et al., 2014). More recently, it has been shown that analyses and short-term forecasts tend to underestimate the peak jet stream wind, the vertical wind shear and the abruptness of the change in wind shear across the tropopause (Schäfler et al., 2020). This calls for a better understanding of processes controlling PV gradients.

Warm conveyor belt (WCB) outflows are one of the main perturbations to the midlatitude waveguide. WCBs are usually poleward and upward, coherent airstreams associated with extratropical cyclones (Wernli and Davies, 1997). Rising slowly with rates not exceeding $50\,\mathrm{hPa\,h^{-1}}$, the warm and moist air in WCBs cools and condenses to form a wide, elongated band of clouds where heavy precipitation and strong surface winds occur (Browning, 1999). During ascents, a large amount of latent heating is released by cloud processes, which representation is considered a major source of uncertainty (Chagnon et al., 2013; Martínez-Alvarado et al., 2014; Joos and Forbes, 2016). This can be explained from the PV perspective, where PV is produced below the level of maximum heating and reduced above (Hoskins et al., 1985). In WCBs, vertical PV dipoles are created with positive PV anomalies in the lower layers and negative PV anomalies in the upper layers (Wernli and Davies, 1997). The negative PV anomalies are then advected into the upper-level ridge by the WCB outflow thus impact the jet stream and the PV gradient at the tropopause. Thus, errors in the PV change by cloud processes lead to errors at upper levels.

Recent studies have shown the presence of mesoscale negative PV structures in WCBs (Harvey et al., 2020; Oertel et al., 2020; Blanchard et al., 2020). Harvey et al. (2020) developed a theory explaining that diabatic heating in the presence of vertical wind shear results in negative PV values on the equatorward side of the jet stream. Oertel et al. (2020) showed with a composite analysis that convective ascents produce horizontal PV dipoles, which persist for about $10\,\mathrm{h}$ and merge to form elongated negative PV bands that can locally accelerate the jet stream. Blanchard et al. (2020) showed that, among three types of organized convection they found in a WCB, only mid-level convection is associated with coherent negative PV bands. These studies further suggest that the mesoscale negative PV structures may accelerate the jet stream locally and potentially influence the downstream circulation.

This paper is focused on the WCB outflow of the Stalactite cyclone observed during the North Atlantic Waveguide and Downstream Impact Experiment (NAWDEX; Schäfler et al., 2018). On 2 October 2016 the WCB outflow was sampled with airborne instruments with the objective to characterize its role in the building of the downstream ridge. Two days later, this ridge became a block over Scandinavia and persisted for several weeks. Previous studies showed the major role of diabatic heating in the Stalactite cyclone's WCB on the subsequent onset of blocking (Maddison et al., 2019, 2020; Steinfeld et al., 2020). Maddison et al. (2019) conducted an ensemble sensitivity analysis in which the Stalactite cyclone is clearly identified





as the main feature influencing the block onset 3–4 days ahead. Maddison et al. (2020) showed, through several sensitivity experiments to a convective parameterization in a global model, that stronger latent heating in the WCB leads to a more amplified ridge after 6 days lead time. Steinfeld et al. (2020) found a strong influence of latent heating in the Stalactite cyclone on the ridge building after 2 days of simulations.

The objective of this study is to examine the WCB outflow at fine scale and to investigate the cloud diabatic effects in the

WCB during a relative short 12-h window. To achieve this objective, we use the convection-permitting simulation described in Blanchard et al. (2020) and run a second simulation set up in the same manner, except with the diabatic impact of clouds turned off. We compare both simulations with airborne Doppler radar and dropsonde measurements taken in the WCB outflow. After showing the cloud diabatic effects in the northwestern edge of the ridge, we trace them back to mid-level convection that occurs in the western flank of the WCB a few hours earlier.

The paper is structured as follows: Section 2 briefly introduces the observations as well as the model simulations and numerical tools used for the analysis. Section 3 describes in detail the airborne observations of the ridge and WCB outflow. Section 4 characterizes the ascents ending in the WCB outflow by studying their Lagrangian back-trajectories, while distinguishing between those with an anticyclonic and a cyclonic curvature. Section 5 focuses on the origin of PV structures in the observed regions before discussing the link with mid-level convective ascents within the WCB. Section 6 concludes the paper.

## 2   Data and methods

### 2.1   Airborne observations

RASTA (RAdar Airborne System; Delanoë et al., 2013) and dropsonde observations were acquired from the SAFIRE (Service des Avions Français Instrumentés pour la Recherche en Environnement) Falcon 20 based on Keflavik, Iceland. On the morning of 2 October 2016, the Falcon 20 flew toward Greenland with the objective of studying the tropopause structure and the WCB

outflow from the Stalactite cyclone (flight 6, Schäfler et al., 2018, see the track in Fig. 1a.). During its cruise, the aircraft flew at around 10 km altitude. On its way back to Iceland, four Vaisala RD94 dropsondes were launched at 10:26, 10:32, 10:36 and 10:41 UTC. In the following, we will discuss the profiles of wind speed, potential temperature and relative humidity (with respect to liquid water below the melting level and to ice above) obtained from the dropsondes as well as the reflectivity and wind speed obtained from the cloud radar RASTA between 10:25 and 11:27 UTC, hereinafter referred to as the 11:00 UTC leg.

The reader is referred to Blanchard et al. (2020) and the references therein for more details on RASTA operated on 2 October.

### 2.2   Meso-NH convection-permitting simulations

Two simulations, REF and NODIA, were performed with the version 5.3 of the non-hydrostatic mesoscale atmospheric Meso-NH model (Lac et al., 2018) over the domain shown in Fig. 1. Both simulations are convection permitting with a grid spacing of 2.5 km horizontally and vertically from 60 m near the surface to 600 m in the upper levels. They are run from 00:00 UTC,

2 October 2016 to 12:00 UTC, 3 October with hourly outputs and initial and boundary conditions provided by the ECMWF





(European Centre for Medium-Range Weather Forecasts) operational analyses. Both simulations share the same parameterizations, differing only in that the heat exchanges in the microphysical scheme are set to zero in NODIA. The REF simulation is described in Blanchard et al. (2020), where more details are given on the parameterizations as well as on radiative tools used to emulate the RASTA and Meteosat Second Generation satellite (MSG) observations.

## 2.3 Lagrangian trajectory and clustering tools

Lagrangian trajectories are calculated online by defining three 3-D passive tracers at each grid point in the simulation domain (Gheusi and Stein, 2002). These tracers are advected by the piecewise parabolic method scheme (Colella and Woodward, 1984), known to conserve well the mass properties of the tracers with a weak numerical diffusion. Back-trajectories are reconstructed from the tracers and are studied for the period from 00:00 until 12:00 UTC on 2 October. This relatively short time window is chosen to ensure that all relevant trajectories remain in the simulation domain during the 12 h period. As in Blanchard et al. (2020), trajectories rising by at least 150 hPa in 12 h are defined as ascents. This threshold is based on the usual criterion of 600 hPa in 48 h used to identify WCB trajectories (e.g., Wernli and Davies, 1997; Martínez-Alvarado et al., 2014; Oertel et al., 2020) but without any constraint on the initial altitude of the trajectories, in contrast with previous studies. Selected ascents thus do not perform a full ascent from the boundary layer to the upper troposphere.

Coherent structures within the WCB are detected with the clustering tool created and implemented in Meso-NH by Dauhut et al. (2016). Coherent updraft structures consist of 3-D objects made of connected grid points for which the vertical velocity is higher than a threshold of $0.3\,\mathrm{m\,s^{-1}}$ as in Blanchard et al. (2020). In the same way, coherent negative PV structures are defined as areas of connected grid points with PV values lower than $-1\,\mathrm{PVU}$ ($1\,\mathrm{PVU} = 10^{-6}\,\mathrm{K\,kg^{-1}\,m^2\,s^{-1}}$).

## 3 Observations of the upper-level ridge at 11:00 UTC

### 3.1 Overview

At 11:00 UTC, the Stalactite cyclone approached Iceland as shown by the infrared MSG brightness temperature (BT, Fig. 1a). The elongated band of primarily high clouds observed east of the simulation domain (BT values less than $-35°$ C) locates the WCB ascent region. High clouds are also present north and partly southwest of the domain and indicate the WCB outflow and cloud head regions, respectively. Mid-level clouds are also detected in these regions (BT values between $-35°$ and $0°$ C). Positive BT values locate the dry intrusion between the cloud head and WCB ascent regions. Some isolated low-level clouds are observed below the dry intrusion. The aircraft crossed the WCB outflow region when flying back to Keflavik during the 11:00 UTC leg.

In REF, the position of the main cloud structures is correctly reproduced although high clouds are more spatially extended in the cloud head and WCB regions (Fig. 1b). The smoothed 2 PVU contour at the 320 K level (blue line) shows that the upper-level ridge, defined as the low PV region, covers the northeast three quarters of the domain. It also highlights a complex





PV structure over the cloud head and along the Greenland coast. North of 60° N, a tongue of high PV value with relatively lower cloud tops cuts the northwestern edge of the ridge in two parts.

In NODIA, the main cloud structures are also reproduced but with higher BT values than in the MSG observation and REF (Fig. 1c). Cloud tops are therefore expected to be lower in these regions. Note that the dry intrusion extends less to the
northwest. The 2 PVU contour shows a pattern similar to REF, but with the PV tongue shifted eastward and less small-scale structures.

The location of the mean sea level pressure (MSLP) minimum of the Stalactite cyclone, represented by the red dotted lines during the simulated 36 h period, shows that the cyclone moves northward on the morning of 2 October. In the ECMWF analysis (Fig. 1a), an abrupt eastward shift then occurs between 12:00 UTC, 2 October and 00:00 UTC, 3 October as the cyclone deepens
and finally moves northwestward towards the Greenland Plateau. REF reproduces well the track of the Stalactite cyclone, including the abrupt eastward shift, as well as its deepening from 968 to around 955 hPa (Fig. 1b). In NODIA, the MSLP minimum values are higher by ∼5 hPa compared to ECMWF and REF (Fig. 1c). The abrupt eastward shift is not reproduced resulting in a more meridian cyclone track. The creation of a second MSLP center to the east therefore has a diabatic origin (see the second MSLP center at 16:00 UTC in Fig. 1b in Blanchard et al. (2020)).

## 3.2 Vertical structure of the upper-level ridge across the flight leg

In the following, we focus on the WCB outflow region overflown by the Falcon 20 aircraft along the 11:00 UTC leg. Its track is indicated by the black lines in Fig. 1a, while the location of the dropsondes launched during the flight are marked by white stars. The observations of the RASTA radar and the dropsondes, combined with the REF results, provide a fine-scale description of the upper-level dynamics in the region.

The vertical structure of reflectivity as seen by RASTA shows a large cloud system between 40° and 27° W (Fig. 2a). Weak reflectivity values (about −20 dBZ) are measured above ≈7 km altitude. These values are characteristic of cirrus-type clouds. Their location is consistent with strong negative BT values shown in Fig. 1a. Reflectivity values then increase below z≈7 km. Reflectivity values of 10 dBZ are measured in the first kilometer of altitude with local peaks greater than 15 dBZ at z≈1 km highlighting the melting level. They are lower between 1 and ≈7 km altitude with local peaks of 10 dBZ. The slope
in the vertical structure of reflectivity between 40° and 35.5° W reveals the warm front associated with the cyclone. Isolated convective structures, highlighted by the reflectivity values greater than 15 dBZ, are present below the slope of the warm front around 39° and 37° W as well as above the Greenland Plateau around 42° W. The lack of radar signal (lower than −20 dBZ) between the warm front slope and the isolated convective structures suggests the presence of a dry air mass. This dry air mass in the mid levels is hardly detectable on the BT field in Fig. 1a.

The WCB outflow region, the slope of the warm front, the isolated convective structures and the dry air mass are well reproduced by REF with reflectivity values similar to those observed (Fig. 2b). The 2 PVU contour shows the PV tongue between 37°–38° W which penetrates the troposphere down to z≈8 km. It also reveals a tropopause fold west of 40° W until z≈6 km, which covers the upper part of the dry air mass. Note the spots of negative PV at the tropopause on the eastern part. In NODIA, the vertical structure of reflectivity shows higher values around the melting level and mid levels compared





to the RASTA observation and REF (Fig. 2c). This can be explained by higher contents of frozen hydrometeors (graupel and snow at the melting level and mid levels, respectively) due to the cut-off of diabatic heating from cloud processes. The upper levels are also impacted. The level of the cloud top on the eastern part does not exceed $\approx 7\,\mathrm{km}$ altitude while it is higher than z=8 km in the observation and REF. Moreover, the tropopause fold and the PV tongue are shifted eastward compared to those simulated in REF. Finally, the dry air mass is not reproduced in NODIA. This can be explained by a lower evaporation of
frozen hydrometeors under the warm slope due to the cut-off of diabatic cooling from cloud processes.

The vertical structure of the horizontal wind speed measured by RASTA shows in part the jet stream with values greater than $25\,\mathrm{m\,s^{-1}}$ above z$\approx 7$ km (in yellow in Fig. 3a). Local peaks of $40\,\mathrm{m\,s^{-1}}$ are measured in the upper levels (in red). The horizontal wind speed decreases below. It is quite homogeneous in the middle and low levels (around $10\,\mathrm{m\,s^{-1}}$) except on the eastern part where it reaches $20\,\mathrm{m\,s^{-1}}$ between 2 and 6 km altitude. Horizontal wind speed values greater than $25\,\mathrm{m\,s^{-1}}$ below
z=2 km around 42° W and 39° W show the presence of a low-level jet along the Greenland coast.

The vertical structure of the horizontal wind speed in the WCB outflow region is well reproduced by REF with horizontal wind speed values close to those measured (Fig. 3b). The simulation completes the description of the jet stream and reveals two intensity maxima, hereafter called jet stream cores. The first is located at z$\approx 8$ km between 43° and 40° W and the second at z$\approx 9$ km between 37° and 31° W. The value of the horizontal wind speed in these two cores locally exceeds $40\,\mathrm{m\,s^{-1}}$ (in
red). The low-level jet is also well reproduced in REF. The black dots show the position of the selected ascents in the cross section at 11:00 UTC. A large number ascents are located above the Greenland Plateau and the low-level jet. Many ascents are also located in the cloudy area, mainly in the eastern part. They are separated in two distinct layers. Most are located in the upper layers, between $\approx 4$ and 10 km altitude, within regions of large wind speed. They feed the eastern jet stream core. The other ascents are located in the lower layers, below z=4 km altitude, near regions of high reflectivity. In NODIA, the jet stream
and the low-level jet are both less intense (Fig. 3c). The western jet stream core is less spatially extended while the eastern jet stream core is shifted eastward. Thus, cloud diabatic processes strengthen the jet stream and modify its location in this case. In NODIA, only the ascents above the Greenland Plateau and the low-level jet are present. They are not studied afterwards in order to focus on the ascents of diabatic origin in REF.

### 3.3 Analysis of the western jet stream core

The vertical profiles of horizontal wind speed, potential temperature ($\theta$) and relative humidity (RH), measured by the four dropsondes launched along the 11:00 UTC leg (see Fig. 1a), are shown in Fig. 4a–d, Fig. 4e–h and Fig. 4i–l, respectively (black lines). The two westernmost dropsondes (at 43.3° and 41.8° W) were launched over the Greenland Plateau, so their profile stop at an altitude close to 2 km. The other two dropsondes (at 40.7° and 39.2° W) were launched along the Greenland coast, over the western edge of the cloudy area. The profiles from REF and NODIA are superimposed on those observed.

The horizontal wind speed profile measured at 43.3° W shows a peak of $\approx 35\,\mathrm{m\,s^{-1}}$ at z=8 km (black line in Fig.4a). At 41.8° and 40.7° W, the horizontal wind speed reaches $42\,\mathrm{m\,s^{-1}}$ and extends vertically from 8 to 10 km altitude (black lines in Fig.4b,c). At 39.2° W, it peaks again at $35\,\mathrm{m\,s^{-1}}$ at these heights (black line in Fig.4d). This zonal variation validates the existence of the western jet stream core seen in Fig. 3b. Its height and intensity are well reproduced by REF (red lines in



Fig.4a–d) but it is slower by $\approx 10\,\mathrm{m\,s^{-1}}$ in NODIA (orange lines in Fig.4a–d). Below the jet stream, the horizontal wind speed decreases down to z$\approx$7 km at the western dropsonde location and z$\approx$5 km at the eastern dropsonde location, in both observation and simulations. The horizontal wind speed then varies from 5 to $20\,\mathrm{m\,s^{-1}}$ until z$\approx$2 km. A second peak of horizontal wind speed of $25\,\mathrm{m\,s^{-1}}$ is measured in the lower troposphere by the two easternmost dropsondes (around z=2 km in Fig. 4c and z=1 km in Fig. 4d). This corresponds to the presence of the low-level jet described in Fig. 3a. The low-level jet is also well reproduced in the two simulations (red and orange lines in Fig. 4c,d) albeit one kilometer lower.

The measured $\theta$ profiles show a slight increase with altitude from $\approx$280 K in the lower levels to $\approx$300 K at z=6 km (black lines in Fig. 4e–h). At 43.3° W, $\theta$ increases sharply above to reach 325 K at z=9 km (Fig. 4e). This layer of high increase in $\theta$ corresponds to the location of the tropopause fold. This is well reproduced by the simulations (red and orange lines in Fig. 4e). At 41.8° and 40.7° W, $\theta$ slightly increases from z$\approx$7 km before increasing abruptly again at z$\approx$9.5 km, both in observations and simulations (Fig. 4f,g). This indicates the presence of a second tropopause level, in addition to the one located at z$\approx$6 km. This is consistent with the locations of the simulated stratospheric PV values and the dynamical tropopause height at the location of the dropsondes. At 39.2° W, $\theta$ increases slightly up to 330 K at z$\approx$9.5 km before increasing suddenly above (Fig. 4h). This altitude corresponds to the dynamical tropopause height at the location of the dropsonde and is also reproduced by the simulations.

The RH profile at 43.3° W shows values of less than 20% above 7 km altitude in both observation and simulations (Fig. 4i). This confirms the absence of high clouds on the western edge of the cross-section. Below, RH reaches larger values, up to 100% and more, at z=4 km. This highlights the location of mid-level clouds over the Greenland Plateau. The measured supersaturation is not reproduced by the simulations because of the saturation adjustment in the microphysical scheme. The RH profiles of the other three dropsondes show high values (close to 100%) above z=7 km (Fig. 4j,k,l). They correspond to the cirrus-type clouds observed in Fig. 2a. A sharp decrease in RH (from 100 to 20%) is measured between $\approx$5<z<7 km, $\approx$4<z<7 km and $\approx$3<z<6 km at 41.8°, 40.7° and 39.2° W, respectively. This is consistent with the location of the dry air mass observed by RASTA and simulated in REF. This decrease in RH is not reproduced in NODIA, which matches the absence of dry air mass in Fig. 2c. Below the dry air mass, the RH show values close to 100%, referring to isolated convective structures along the Greenland coast in Fig. 2a,b. Overall, the measured vertical profiles complement the RASTA observations and are consistent with the vertical structures simulated in REF.

## 4  Evolution of ascents in the WCB outflow

### 4.1  Selection of ascents

The location of air parcels respecting the ascent criterion of 150 hPa between 00:00 and 12:00 UTC, 2 October is shown at 11:00 UTC for REF (Fig. 5a) and NODIA (Fig. 5b). The colored contours represent the equivalent potential temperature $\theta_e$ at z$\approx$1 km altitude at 11:00 UTC.

In REF, three regions of high ascent frequency are highlighted (in blue and green in Fig. 5a). The first region is located north of the cyclone center, above high $\theta_e$ values. It corresponds to the WCB outflow region overflown by the aircraft. The red box is





used as a mask to select the ascents located there at 11:00 UTC. The second region is located southwest of the domain, between 53°–61° N and 40°–25° W. It is associated with the cloud head region. The tightening of iso-$\theta_e$ contours in this region shows the winding of the bent-back front around the cyclonic center, where some local peaks of high ascent frequency are located. Some ascents are identified further westward. The third region is located northwest of the domain, above Greenland.

In NODIA, only two regions of high ascent frequency are highlighted: Greenland and the bent-back front region (Fig. 5b). Thus, ascents in these two regions have a dynamic origin. Those above Greenland are as numerous as in REF. They are produced by the combined effect of the warm front dynamics and orographic forcing caused by the Greenland Plateau. A higher frequency of ascents is even identified compared to REF along the bent-back front, between 54°–56° N and 35°–30° W. In contrast, ascents are almost lacking in the WCB outflow region (red box), which indicates their diabatic origin. In the following, only ascents from this region are further discussed.

### 4.2 Location of the selected ascents

The ascents simulated by REF in the WCB outflow region at 11:00 UTC are now examined. An overview is presented in Fig. 6 by showing a sample of their trajectories colored by altitude between 00:00 UTC and 12:00 UTC. For better visibility, anticyclonic trajectories (Fig. 6a) are distinguished from cyclonic trajectories (Fig. 6b). Cyclonic (anticyclonic) trajectories are identified based on their cyclonic (anticyclonic) curvature between 06:00 UTC and 12:00 UTC.

At 00:00 UTC, most anticyclonic ascents are located along a band extending from ≈56° N and ≈30° W to ≈53° N and ≈22° W (red stars in Fig. 5a). Their position corresponds to the location of the bent-back front at this time (not shown). The majority of cyclonic ascents also starts along this band, while some start further north (red stars in Fig. 5b). At 06:00 UTC, all the ascents have been advected northward by the large-scale flow (black dots in Fig. 5a,b). Most of the anticyclonic ascents end in the eastern part of the 11:00 UTC leg (brown circles in Fig. 5a). A few of them end further north. Some cyclonic ascents also end in the eastern part of the 11:00 UTC leg but the majority end further south (brown circles in Fig. 5b).

The anticyclonic ascents are higher in altitude than the cyclonic ascents. They are located between ≈4000<z<7000 m at 00:00 UTC (in light blue and green in Fig. 5a) and ≈7000<z<10000 m at 12:00 UTC (in orange). In contrast, the cyclonic ascents remain below z≈5000 m between 00:00 UTC and 12:00 UTC (in blue in Fig. 5c). Thus, the anticyclonic ascents correspond to the ascents found at 11:00 UTC in the eastern jet stream core (Fig. 3b) and the cyclonic ascents to those found in the lower layers. The ascents with an anticyclonic curvature are similar to the anticyclonic branch of the WCB (Martínez-Alvarado et al., 2014). They are therefore expected to impact the upper-level ridge in the WCB outflow region.

### 4.3 Properties of the selected ascents

The temporal evolution of altitude and PV along the selected ascents simulated in REF is examined (Fig. 7). The 2 h part of the trajectories which undergo an ascent greater than 100 hPa are also discussed. They are referred to as rapid segments thereafter. Overall, there are about as many anticyclonic ascents (53%) as there are cyclonic ascents (47%).

As expected, anticyclonic ascents are at a higher altitude than cyclonic ascents (Fig. 7a). The interquartile ranges (shaded color) do not overlap. The anticyclonic ascents (in blue) are located at z≈4 km at 00:00 UTC and rise continuously until





z≈7 km at 12:00 UTC, on average. Some exceed z=8 km at the end of the trajectory. Anticyclonic rapid segments are more numerous at the beginning of the trajectories and take place around z=4 km (black boxplots in Fig. 7a). Their number then decrease with time. This suggests a strong mid-level convective activity in the first hours of simulation, close to the region identified at 00:00 UTC in Fig. 6a. The cyclonic ascents (in orange) are located at z≈1 km at 00:00 UTC, on average. Contrary to the anticyclonic ascents, they stay at the same altitude until 04:00 UTC before rising to z≈3 km (on average) at 12:00 UTC.

Some start close to the surface. The cyclonic rapid segments occur later than the anticyclonic rapid segments (red boxplots in Fig. 7a). They are also located at lower altitudes, around z=2 km. This suggests the presence of shallow convective activity at that time.

Potential vorticity decreases slowly along the anticyclonic ascents with PV values ranging from 0.6 PVU at 00:00 UTC to 0.4 PVU at 12:00 UTC on average (Fig. 7b). The interquartile range shows PV values reaching 1.2 PVU at 02:00 UTC and

0.0 PVU during the 12 h period (shaded blue). In contrast, the averaged PV value along the cyclonic ascents first remains around 0.3 PVU then increases between 04:00 UTC and 08:00 UTC when the rapid cyclonic segments occur. As for the anticyclonic ascents, the interquartile range shows PV values between 0.0 PVU and 1.2 PVU (shaded orange). This contrasting PV evolution between upper and lower levels of the troposphere corresponds to the classical view of Wernli and Davies (1997). However, rapid segments, in particular anticyclonic ones, indicate negative PV values from 03:00 UTC onward. This suggests

that convection, especially at mid levels, is associated with negative PV creation. The origin of this process is detailed thereafter.

## 5 Origin of updrafts and negative PV

### 5.1 Negative PV bands at upper levels

The ridge associated with the Stalactite cyclone is examined at 11:00 UTC on maps of PV and horizontal wind at the $\theta$=320 K level in REF and NODIA (Fig. 8a and Fig. 8b, respectively). The same maps are shown at 06:00 UTC (Fig. 8c,d) and at

02:00 UTC (Fig. 8e,f) in order to track the evolution of the ridge back in time in both simulations. PV values larger than 2 PVU at $\theta$=320 K show stratospheric air (in white) and PV values lower than 2 PVU tropospheric air (in colors). The jet stream follows the tropopause where the PV gradient is strongest (red arrows).

At 11:00 UTC, the ridge in REF largely covers the northeastern part of the domain (Fig. 8a). Above the Greenland Plateau, stratospheric air corresponds to the upper part of the tropopause fold, as shown in Fig. 2b. Further east, the PV tongue with

stratospheric air is located between ≈38°–35° W around 64° N. It cuts the northwestern edge of the ridge in two parts where the horizontal wind speeds exceed 45 m s$^{-1}$, corresponding to the two jet stream cores described in Fig. 3b. In the eastern part, elongated negative PV bands (in blue) are simulated between 62°–66° N and 35°–25° W. This region coincides with the location of upper-level anticyclonic ascents (see brown circles in Fig. 6a). A second region with elongated negative PV bands is simulated further south, between 54°–58° N and 22°–15° W, along another jet stream core. This second region was overflown

at 16:00 UTC by the Falcon 20 aircraft and is further described in Blanchard et al. (2020).

In NODIA, the PV tongue is shifted eastward compared to REF (Fig. 8b). The eastern part of the northwestern edge of the ridge is also shifted eastward. The negative PV bands are not reproduced by NODIA, neither in this region nor in the second





region further south. This reveals that the elongated negative PV bands are created by cloud diabatic processes. The wind speed is less intense in the two jet stream cores, as already shown in Fig. 3c. Following the ridge, the jet stream is also less curved to
the west.

At 06:00 UTC, the ridge is located further south in the domain (Fig. 8c,d). Its western part extends until 40° W in both REF and NODIA. But the part to the east of the PV tongue extends further west around 60° N in REF compared to DIA. The elongated negative PV bands in REF are more concentrated there than at 11:00 UTC. Once again, they are not reproduced in NODIA and the jet stream is less intense than in REF. At 02:00 UTC, the ridge does not differ much between the two
simulations (Fig. 8e,f). This is understandable, as this time is close to the initialization of the simulations, and means the cloud diabatic processes have not yet strongly influenced the upper-level dynamics. In particular, negative PV structures are found in both simulations at that time, and are already present in the initial conditions (not shown).

Overall, the comparison between REF and NODIA shows the impact of cloud diabatic processes that occurred in the WCB on the upper-level dynamics. These processes create negative PV bands that persist over time and are found at the northwestern
edge of the ridge at the time of observations. The negative PV bands reinforce the PV gradient at the tropopause level and thus the jet stream.

## 5.2 Origin of the negative PV bands

The origin of the negative PV bands is now investigated. Firstly, rapid segments along anticyclonic ascents are examined at 02:00 UTC in REF (Fig. 9a,b). Their location corresponds to the region of origin of both the anticyclonic ascents (brown circles
in Fig. 6a) and the elongated negative PV bands found in the WCB outflow region (Fig. 8e). Furthermore, time evolutions have shown that anticyclonic rapid segments are most numerous during the early simulation hours (see black boxplots in Fig. 7a). Secondly, the creation of negative PV at upper levels is assessed at the same time by showing the negative PV structures and their altitude (Fig. 9c,d). For easier interpretation, only coherent negative PV structures are discussed here. They are defined as regions with PV values less than $-1$ PVU. The same clustering approach is used to identify the base of coherent updrafts,
defined as regions with vertical wind speed values greater than $0.3\,\mathrm{m\,s^{-1}}$. Finally, the results are compared with those of NODIA to highlight the impact of cloud diabatic processes on upper-level dynamics (Fig. 10).

Most anticyclonic rapid segments in REF are located along the bent-back front, above a region of high $\theta_e$ values at 02:00 UTC (black dots in Fig. 9a). Similarly, coherent updrafts are located along the bent-back front at lower and mid levels. Some anticyclonic rapid segments are also located further southwestward but are less numerous and not discussed here. A meridionally
oriented vertical cross-section illustrates that anticyclonic rapid segments are mainly located between $1\,\mathrm{km}$ and $\approx 4\,\mathrm{km}$ altitude along the bent-back front, where vertical wind speeds from $0.1\,\mathrm{m\,s^{-1}}$ to $0.5\,\mathrm{m\,s^{-1}}$ are simulated (black dots in Fig. 9b). These ascents correspond to the lower-levels updrafts in Fig. 9a and originate from the frontal uprising, highlighted by the tightening of iso-$\theta_e$ contours in the lower layers (black lines in Fig. 9b). Anticyclonic rapids ascents are also identified at higher altitude within two convective cells of vertical velocity greater than $0.9\,\mathrm{m\,s^{-1}}$. The first cell is located between $4\,\mathrm{km}$ and $\approx 7\,\mathrm{km}$ al-
titude around 56.5° N and the second between $4\,\mathrm{km}$ and $\approx 6\,\mathrm{km}$ altitude around 57° N. A third cell of relatively high vertical





wind speed is located between 6 km and ≈9.5 km altitude around 57° N but it does not match the criteria of rapid segments (see Blanchard et al., 2020, for a discussion). These mid-level convective cells correspond to the mid-level updrafts in Fig. 9a.

The top altitude of negative PV structures is shown at 02:00 UTC in Fig. 9c (shading) along with the jet stream (black lines). The location of negative PV structures is consistent with updrafts and anticyclonic rapid segments at 02:00 UTC, which follow

the eastern edge of the jet stream at upper levels and the bent-back front at lower levels. The vertical cross-section reveals the presence of mesoscale horizontal PV dipoles around the first and second mid-level convective cells (Fig. 9d). They are located above a low-level jet and below the upper-level jet stream. The negative PV poles are on the jet stream side and reach values lower than $-2$ PVU, while the positive PV poles reach values larger than $2$ PVU. This description is coherent with the findings of Oertel et al. (2020) and Blanchard et al. (2020). Strong positive PV values are also visible in the low-level jet, below the

anticyclonic rapid segments. This corresponds to the classical view of Wernli and Davies (1997), i.e., a vertical PV dipole with positive anomaly below the maximum level of diabatic heating.

Thus, the anticyclonic rapid segments that end in the WCB outflow region at 11:00 UTC originate mainly from the same region at 02:00 UTC. Some are lifted relatively gently along the bent-back front in the lower layers while others are accelerated upwards within convective cells located in the middle layers. The latter create mesoscale horizontal PV dipoles whose pole

reaches strongly negative values below the jet stream.

The same fields are shown for NODIA, also at 02:00 UTC and in the same region (Fig. 10). The bent-back front is less pronounced and the corresponding updrafts are absent in NODIA (Fig. 10a). Only about ten coherent updrafts are located at upper levels. Moreover, no anticyclonic rapid segments are present. This is consistent with the absence of convective cells in the vertical cross-section and is explained by the greater stability of the middle layers compared to REF, in particular at the

western edge of the cloudy area (Fig. 10b). Negative PV structures are also rare in NODIA (Fig. 10c). Some are present at upper levels, along the eastern edge of the jet stream, and are inherited from the initial conditions. The vertical cross-section illustrates the absence of horizontal PV dipoles in the mid-level troposphere (Fig. 10d). Negative PV values close to the jet stream core (around z=9 km at ≈56.50° N) and positive PV values in the low-level jet are both weaker compared to REF. The impact on the dynamics is contrasted at this time, as the jet stream core is not yet impacted while the low-level jet has less

intense horizontal winds.

To summarize, the comparison between REF and NODIA shows the influence of cloud diabatic processes, which are at the origin of mid-level convective cells within the cloud region. These cells diabatically create horizontal PV dipoles at mid levels with a pole reaching strongly negative values lower than $-2$ PVU. Negative PV structures then persist with time and participate in the westward extension of the ridge and the associated tightening of tropopause PV gradients, which result in the

strengthening of the upper-level jet.

## 6    Conclusions

This paper focuses on the WCB outflow associated with the Stalactite cyclone located close to the Icelandic coast on 2 October 2016. To this end, fine-scale observations of upper-level dynamics in the WCB outflow region were performed using the





RASTA radar during a flight of the Falcon 20 aircraft operated during the NAWDEX field campaign. In addition, in-situ mea-
surements of cloud structure and dynamics were provided by four dropsondes launched from the Falcon 20. The observations
are combined with results from two Meso-NH convection-permitting simulations of the cyclone during its mature phase. The
first is defined as the reference simulation (REF), while in the second simulation (NODIA) heat exchanges from cloud pro-
cesses are set to zero in the microphysical scheme. The main cloud structures observed by the MSG satellite are well simulated
on a kilometer scale in REF, whereas the cloud tops are generally too low in NODIA. Moreover an abrupt eastward shift of the
cyclone's trajectory, according to the ECMWF analysis, is well captured by REF but is not reproduced by NODIA.

RASTA observations show structures with low and high reflectivity values in the upper and lower troposphere, respectively.
They thus highlight the presence of cirrus-type clouds in the WCB outflow region as well as convective activity within the
WCB and above the Greenland Plateau. These observations also reveal the existence of a dry air mass between the warm front
and the Greenland Plateau. The reflectivity structures simulated in REF are in agreement with the observations, while the cloud
tops are lower and the dry air mass is absent in NODIA. Two regions of low dynamical tropopause are found in the simulations,
a PV tongue that cuts the northwestern edge of the ridge in two parts and a tropopause fold at its outer boundary, while local
structures of negative PV are found in the inner part. In NODIA, the dynamical tropopause is lower than in REF, the tropopause
fold and the PV tongue are shifted eastward, and negative PV structures are rarer.

RASTA also measures an increase in horizontal wind speed with altitude, with values locally exceeding $40 \, \mathrm{m \, s^{-1}}$ associated
with the jet stream around z=8 km. A low-level jet is observed along the Greenland coast with horizontal wind speed about
$25 \, \mathrm{m \, s^{-1}}$ below z=2 km. REF completes the measurements by highlighting the presence of two jet stream cores located near
the tropopause fold and near the PV tongue. NODIA also simulates two jet stream cores but with lower intensity and shifted
towards the east. Similarly, the low-level jet is well reproduced in REF but is weaker in NODIA. Dropsonde observations
confirm the existence of the western jet stream core and the tropopause fold found in the simulations. They also agree with the
low-level jet and dry air mass measured by RASTA.

Air parcels undergoing an ascent of at least 150 hPa in 12 h are identified in REF and NODIA with online Lagrangian
trajectories. In REF, three main regions of ascents appear at the time of the observations. They are associated with the WCB
outflow region, the cloud head and the Greenland Plateau. In contrast, only ascents located in the cloud head and above the
Greenland Plateau are simulated in NODIA. They thus have a dynamical origin, due to a combination of orographic forcing
and frontal dynamics, while ascents in the WCB outflow clearly arise from cloud diabatic processes. The focus is on the latter,
which are found in REF along the flight leg and feed the eastern jet stream core.

The ascents that end in the WCB outflow region are further separated between anticyclonic and cyclonic curvatures, which
are approximately equally represented. Most of them start in the same region near the cyclone's bent-back front, the anticy-
clonic ascents being higher than the cyclonic ascents. The anticyclonic ascents end in the northwestern edge of the ridge at high
altitudes and are reminiscent of the anticyclonic branch of the WCB (Martínez-Alvarado et al., 2014). The cyclonic ascents
stay in the lower levels during the 12 h window and end further south. The rapid segments – defined as the portion of the
ascents that rise by at least 100 hPa in 2 h – occur mainly in the middle troposphere along the anticyclonic ascents in the first
hours of simulation and in the lower troposphere along cyclonic ascents later on. The time evolution of PV shows an increase





along cyclonic ascents and a slow decrease along anticyclonic ascents. It is consistent with the vertical dipole of PV anomalies
centered around the level of maximum diabatic heating described in Wernli and Davies (1997) for slantwise ascents. However,
not only low but also negative PV values are reached by rapid, especially anticyclonic segments, suggesting a convective origin
as in Oertel et al. (2020).

In a comparative evolution between REF and NODIA, the northwestern edge of the ridge consistently extends further
west and the corresponding jet stream core is more intense and meandering. This confirms that cloud diabatic processes may
reinforce the ridge and the jet stream in the WCB outflow region, as found in previous studies (e.g., Chagnon et al., 2013; Joos
and Forbes, 2016). Furthermore, elongated negative PV bands are simulated in this region in REF but not in NODIA. Such
negative PV bands were also found by Blanchard et al. (2020) along the flank of a jet stream core for the same case study. Here,
the comparison between REF and NODIA highlights that they are diabatically produced, in agreement with recent studies using
mesoscale simulations and observations (Oertel et al., 2020; Harvey et al., 2020). The negative PV bands originate from the
same region as anticyclonic ascents ending in the WCB outflow region, which leads to further examine their potential link.

During the first hours of the REF simulation, a clustering analysis identifies the updraft objects above the bent-back front,
whose location matches the anticyclonic rapid segments at that time. Negative PV structures are located in the same region
at mid and upper levels. While the identified updrafts in the lower layers are due to frontal dynamics and characterized by a
relatively low vertical velocity of about $0.3\,\mathrm{m\,s^{-1}}$, in the middle levels they take the form of convective cells and reach about
$1\,\mathrm{m\,s^{-1}}$. These cells are located at the western edge of the cloudy area, below the jet stream core and above the low-level jet,
in a region of conditional instability. This description matches with the organized mid-level convection in the WCB region
found by Blanchard et al. (2020). In addition, horizontal PV dipoles are found around the mid-level convective cells with the
negative pole facing the jet stream, which confirms the theory developed in Harvey et al. (2020) and the findings of Oertel et al.
(2020). In contrast, updraft objects and negative PV structures are absent from NODIA, as well as mid-level convective cells
and horizontal PV dipoles.

Overall, the results highlight that negative PV structures are diabatically created by mid-level convection. These structures
are then advected by the upper-level anticyclonic flow into the northwestern edge of the ridge, where they persist for about
$10\,\mathrm{h}$ before dispersing. During this time, they participate in extending the ridge westward, strengthening PV gradients at the
tropopause level and intensifying the jet stream. The results thus suggest that mid-level convection contributes to ridge building
and questions its role in large-scale dynamics. As parameterization schemes often struggle to represent updrafts that do not
start in the boundary layer, the representation of mid-level convection may be a source of uncertainty for the prediction of the
downstream atmospheric circulation in global models.

*Data availability.*    All data are available from the authors upon request.



*Author contributions.* NB performed the simulations and the analyses under the supervision of FP and JPC, JD provided the observations,
and all authors prepared the manuscript.

*Competing interests.* The authors declare that they have no conflict of interest.

*Acknowledgements.* Computer resources were allocated by GENCI through Project 90569. The research leading to these results has received
funding from the ANR-17-CE01-0010 DIP-NAWDEX project. The SAFIRE Falcon contribution to NAWDEX received direct funding from
L'Institut Pierre-Simon Laplace (IPSL), Météo-France, Institut National des Sciences de l'Univers (INSU) via the LEFE program, EUFAR
Norwegian Mesoscale Ensemble and Atmospheric River Experiment (NEAREX), and ESA (EPATAN, Contract 4000119015/16/NL/CT/gp).



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



**Figure 1.** 10.8 $\mu$m brightness temperature (in °C) at 11:00 UTC, 2 October 2016 (a) observed by MSG and simulated by (b) REF and (c) NODIA. In (a), (b) and (c), the cyclone track and value of the MSLP minimum are shown (red dotted line, red mark every 3 h) for the ECMWF analysis, REF and NODIA, respectively. The MSLP minimum is tracked every 6 h within a radius of 250 km from its prior position in the ECMWF analysis and every 1 h within a radius of 160 km in the simulations. In (a), the black line shows the track of the Falcon 20 aircraft and the 11:00 UTC leg whereas the white stars show the location of the dropsondes shown in Fig. 4. In (b) and (c), MSLP is shown with white contours every 4 hPa between 964 and 1016 hPa and the smoothed 2 PVU at 320 K with blue contours.

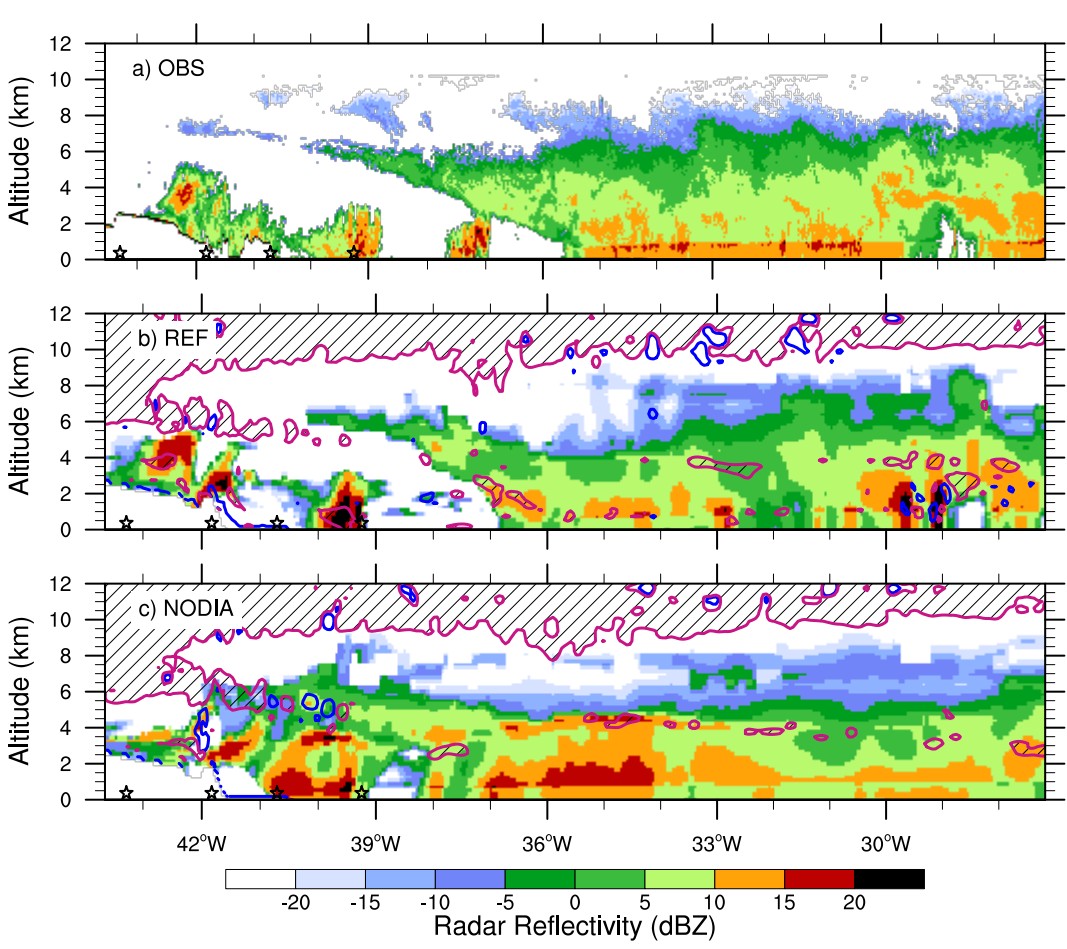

**Figure 2.** Reflectivity (in dBZ) (a) measured by RASTA and simulated by (b) REF and (c) NODIA along the 11:00 UTC leg (black line in Fig. 1). The black stars show the position of the dropsondes shown in Fig. 4. In (b) and (c), magenta and navy blue contours show PV values equal to 2 PVU and −1 PVU, respectively, with hatching for PV values greater than 2 PVU.



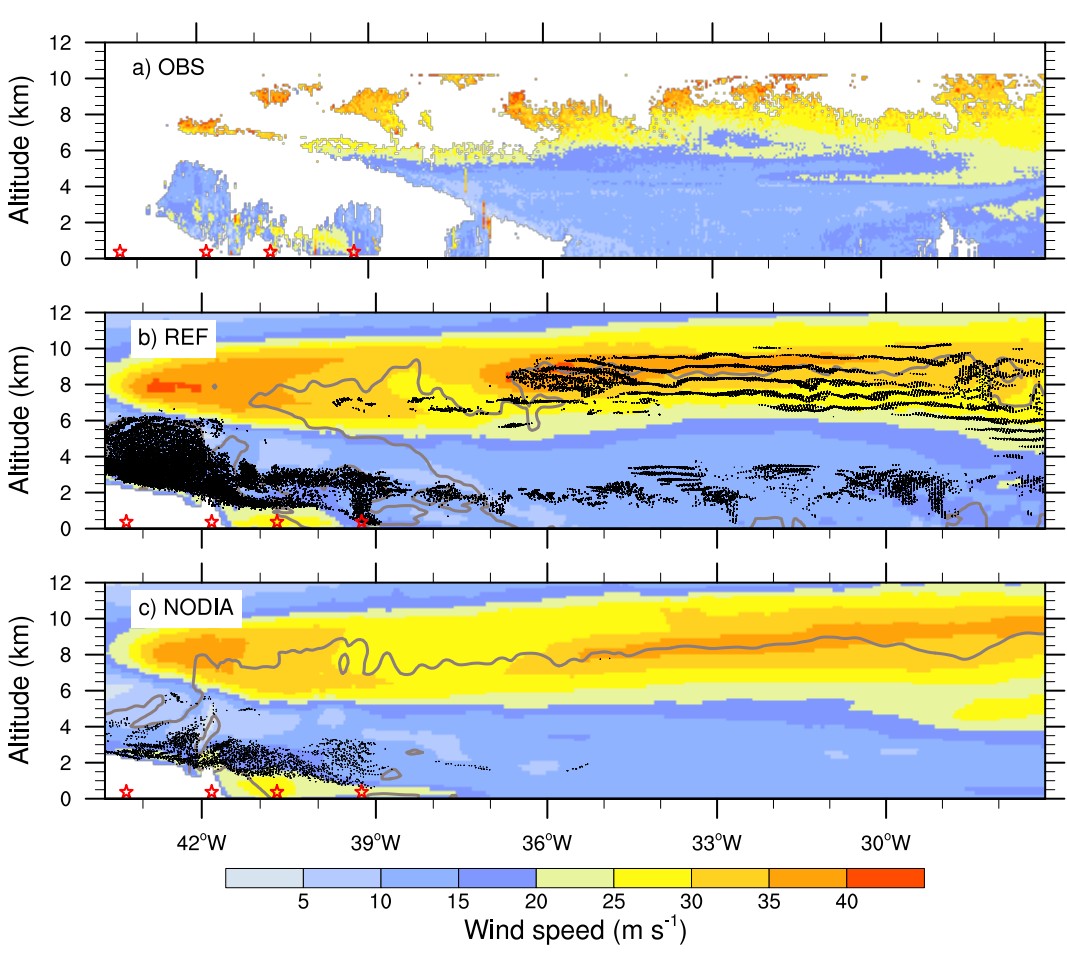

**Figure 3.** Horizontal wind speed (in m s$^{-1}$) (a) measured by RASTA and simulated by (b) REF and (c) NODIA along the 11:00 UTC leg (black line in Fig. 1). The red stars indicate the position of the dropsondes shown in Fig. 4. In (b) and (c), the black dots indicate the position of the selected ascents (see text for details) and the grey lines show the condensed water content equal to 0.02 g kg$^{-1}$.



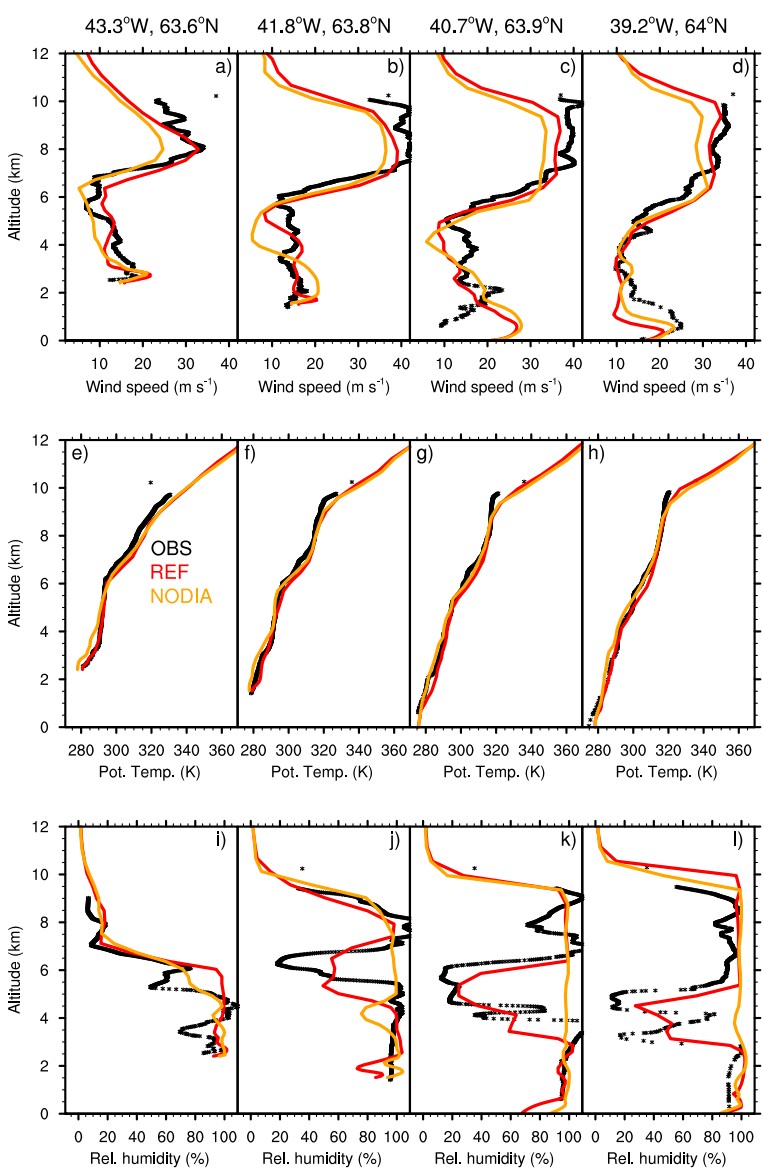

**Figure 4.** Profiles of (a–d) wind speed, (e–h) potential temperature and (i–l) relative humidity at (a, e, f) 43.3° W, 63.6° N, (b, f, j) 41.8°W, 63.8° N, (c, g, k) 40.7° W, 63.9° N and (d, h, l) 39.2° W, 64° N launched at 10:26, 10:32, 10:36 and 10:41 UTC, respectively. Their location is shown as white stars in Fig. 1a.



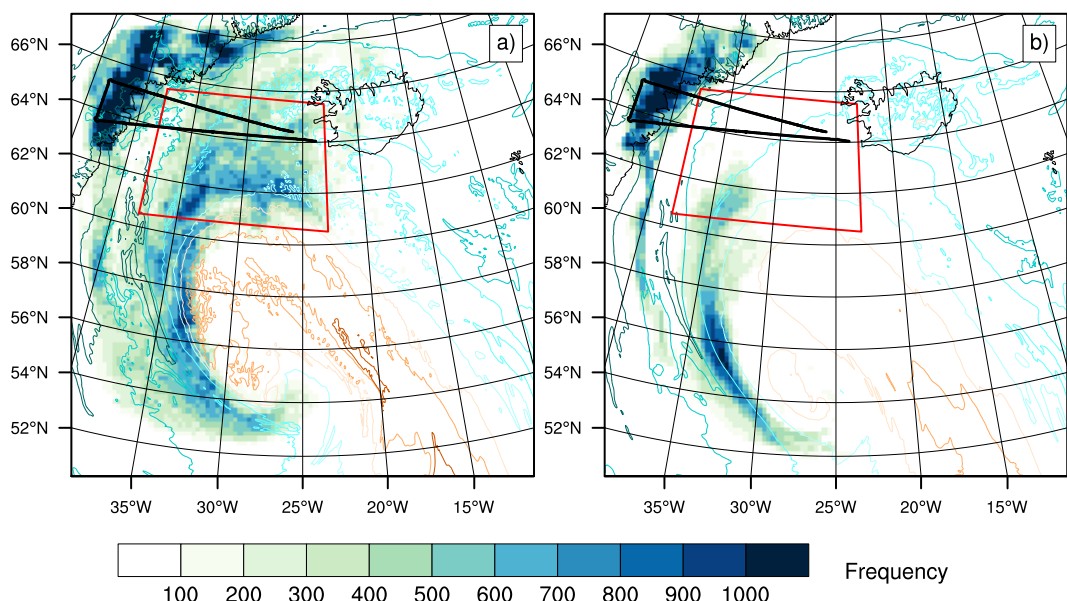

**Figure 5.** Spatial frequency of air parcels belonging to the ascents fulfilling the ascent criterion (shading) and $\theta_e$ at 1 km altitude (colored lines every 4 K between 288 and 312 K) at 11:00 UTC simulated by (a) REF and (b) NODIA. The black line shows the track of the Falcon 20 aircraft and the red box the region where the ascents are selected at 11:00 UTC.



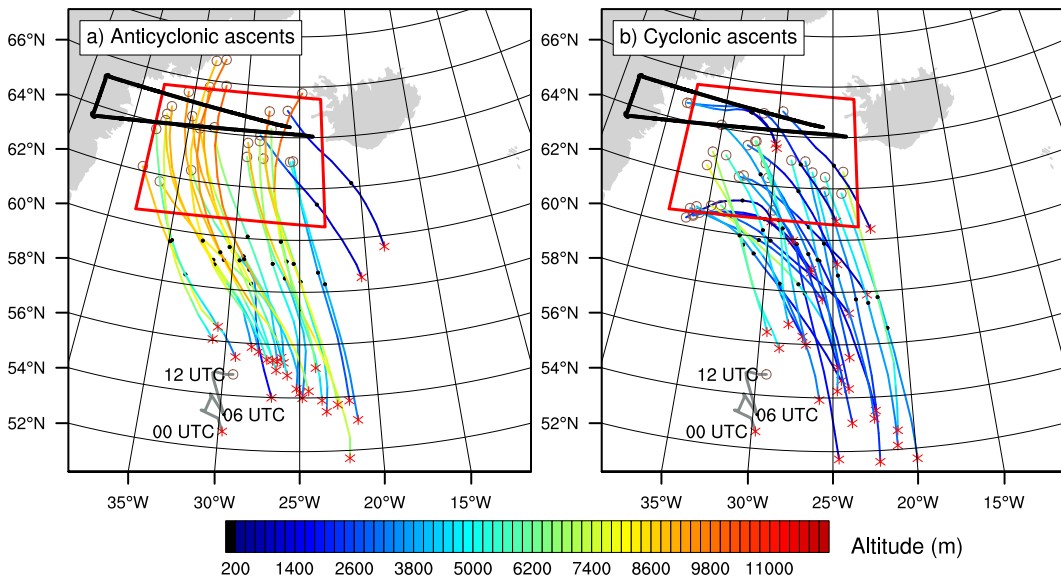

**Figure 6.** Selected trajectories colored by altitude between 00:00 and 12:00 UTC for (a) anticyclonic ascents and (b) cyclonic ascents simulated by REF. Only 40 trajectories are plotted for each category of ascents. A red cross, a black dot and a brown circle show the location of each trajectory at 00:00, 06:00 and 12:00 UTC, respectively. The black line shows the track of the Falcon 20 aircraft, the grey line the position of the MSLP minimum and the red box the region where the ascents are selected at 11:00 UTC.

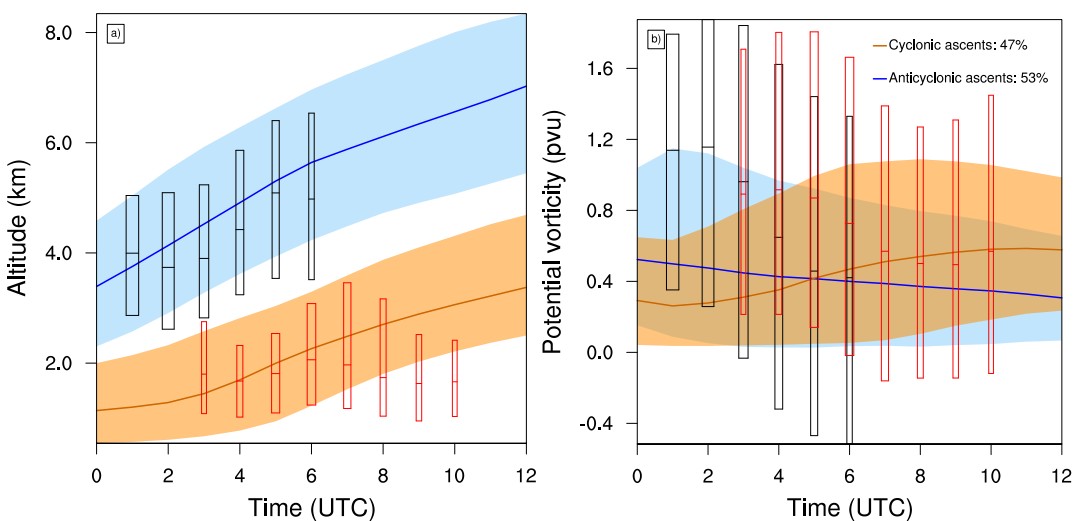

**Figure 7.** Time evolution of (a) altitude (in km) and (b) PV (in PVU) between 00:00 and 12:00 UTC along the selected trajectories in REF. The median (colored bold line) and the 25th-75th percentiles (shaded color) are shown for cyclonic (orange) and anticyclonic (blue) ascents. The median and the 25th-75th percentiles are shown with boxplots for the 2 h rapid cyclonic (red) and anticyclonic (black) segments. Boxplots are displayed only where the number of rapid segments lies above average and their width is scaled with this number.

**Figure 8.** PV (shading) and wind above 45 m s$^{-1}$ (red arrows) on the 320 K level at (a, b) 11:00 UTC, (c, d) 06:00 UTC and (e, f) 02:00 UTC simulated by (a, c, e) REF and (b, d, f) NODIA. The black line in (a, b) shows the track of the Falcon 20 aircraft and the box in (e, f) indicates the zoom in Figs. 9 and 10.



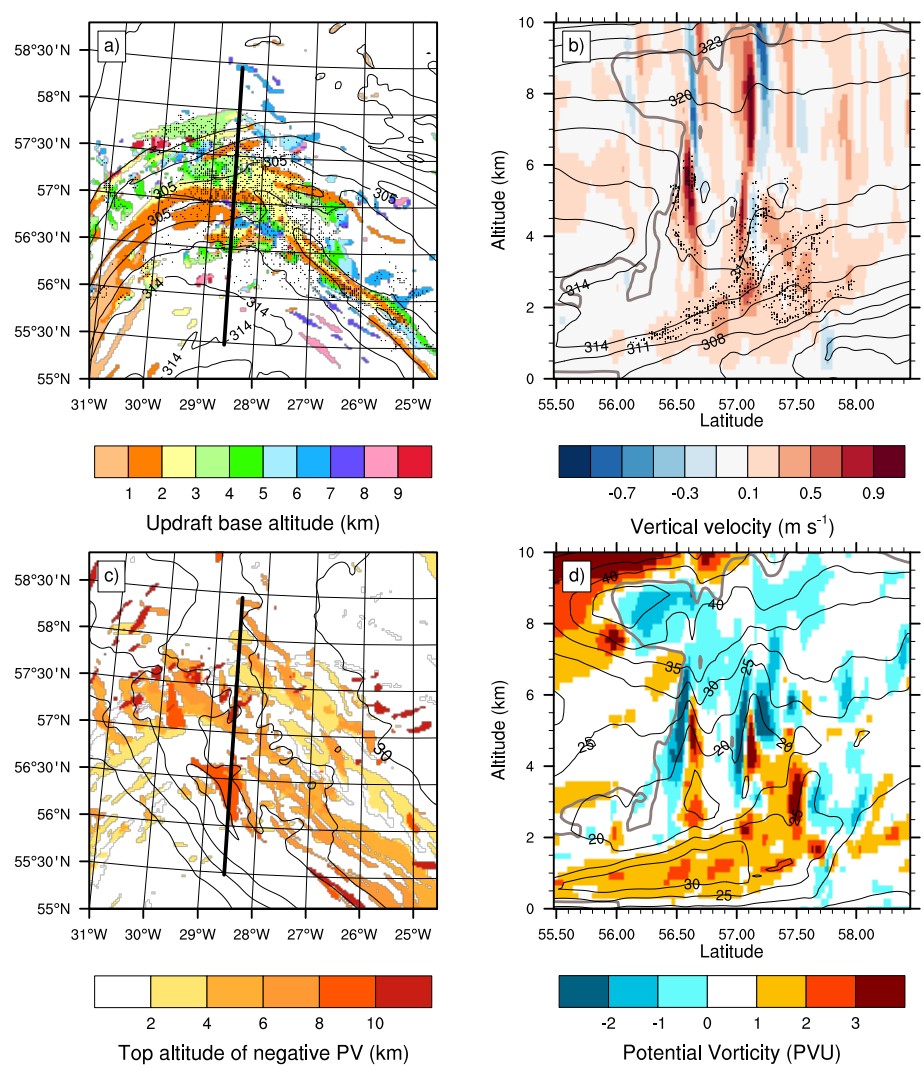

**Figure 9.** Updrafts and potential vorticity at 02:00 UTC in the REF simulation: (a) base altitude of updrafts (shading, in km) and $\theta_e$ at 1 km height (black contours every 3 K); (b) vertical velocity (shading, in m s$^{-1}$), $\theta_e$ (black contours every 3 K) and cloud variables (thick grey contour above 0.1 g kg$^{-1}$) along the vertical cross-section illustrated in (a); (c) top altitude of negative PV structures (shading, in km) and horizontal wind speed on the 320 K level (contours every 5 K above 30 m s$^{-1}$); (d) PV (shading, in PVU), horizontal wind speed (black contours every 5 K) and cloud variables (thick grey contour above 0.1 g kg$^{-1}$) along the vertical cross-section illustrated in (b). Dots in (a, b) indicate the location of rapid anticyclonic segments, reduced to one every 10 in (a).

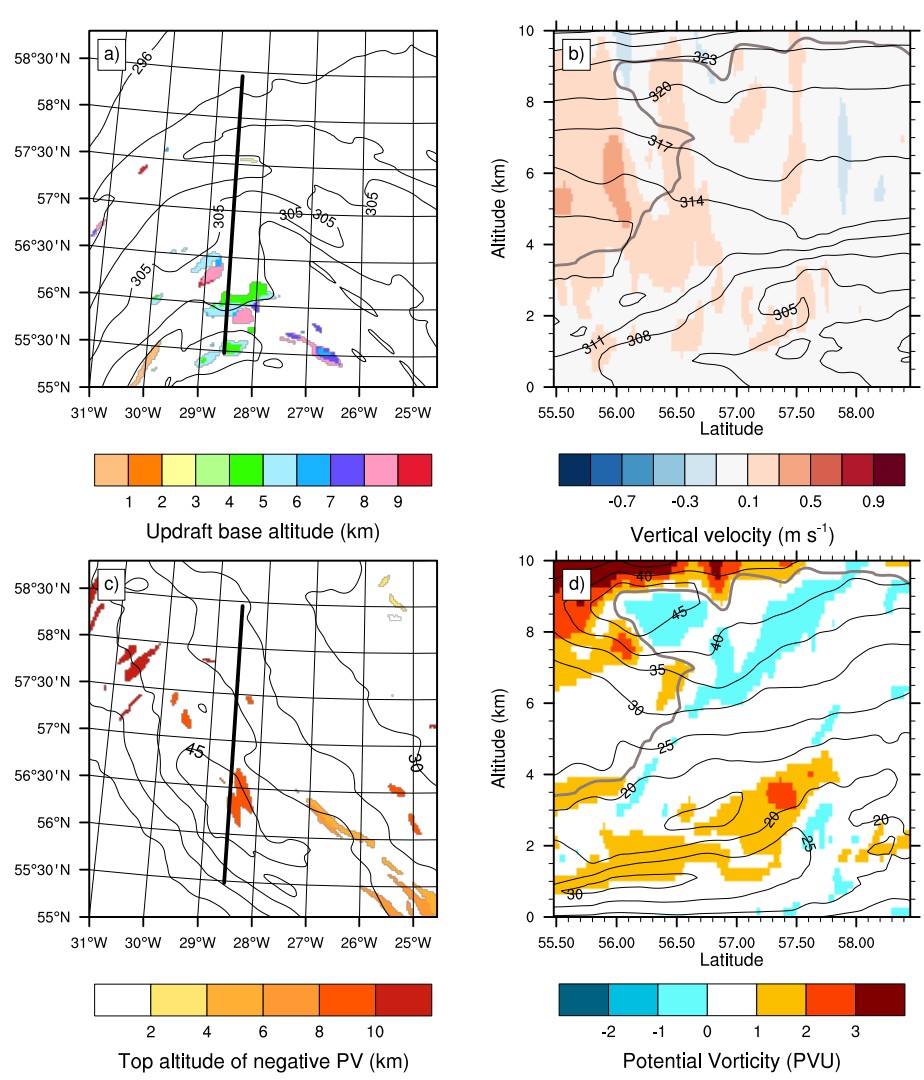

**Figure 10.** As in Fig. 9 but for the NODIA simulation.