# Peer review of "Mid-level convection in a warm conveyor belt accelerates the jet stream"

_Weather and Climate Dynamics, 2020_

## Referee Comment (RC2) · Anonymous Referee #2 · 25 Nov 2020

Blanchard et al. present a detailed analysis of convection embedded in a WCB and how this affects the upper-tropospheric flow. The study is based on observations taken during the North Atlantic Waveguide and Downstream Impact Experiment and convection-permitting simulations. A reference simulation (REF) generally agrees with the observations and represents key features such as the WCB outflow, a dry region below this outflow and the cloud head associated with the bent-back warm front. A second simulation is performed with latent heating exchanges due to cloud processes being turned off (NODIA). A comparison of the two simulations reveals that elongated bands of negative PV are missing the the NODIA simulation pointing to their diabatic origin. Indeed, the analysis of trajectories and vertical cross section through the WCB suggests that mid-level convection embedded in the WCB is responsible for generating

the bands of negative PV in a vertically sheared environment. This is in line with recent studies by Harvey et al. (2020) and Oertel et al. (2020). The study is well written, the figures are mostly clear and the methods are sound. As the paper confirms recent research using novel observations and a slightly different approach (simulations with latent heat release switched on/off), I recommend the article to be published in WCDD after the following comments have been addressed.

**Broad comments**

1) The REF and NODIA simulations are compared qualitatively throughout the paper. To my impression it would be helpful if the authors provided quantitative estimates of the differences between the simulations since it is sometimes difficult to spot the differences by eye. As an alternative, difference plots would help the reader to fully appreciate the differences (e.g, Fig. 3, 8) which are discussed in the text.

2) The individual subsections are quite often introduced by describing what is shown in the figures. These descriptions are not necessary since they are also provided in the figure captions. Instead, it would be helpful if the authors described the purpose of each subsection in one to two sentences. This would help to guide the reader through the manuscript.

**Minor comments**

l. 2: Please clarify that "their" is referring to WCBs and not to "ridges".

l. 9: Since the "mesoscale structures" are mentioned here for the first time. Please specify what the "mesoscale structures" are. Are these the tropopause fold and the jet stream core?

l. 22: Also PV gradients along zonal flows form a waveguide. Please include this as well.

l. 32: I'd suggest to also cite at least one of the early studies, e.g., by Browning et al. (1973) and Harrold (1973).

l. 32: Other studies state that WCBs are characterized by "rapid ascent" (e.g., Eckardt et al. 2004). Compared to deep convection the WCB ascent may be considered as "slow". Perhaps specify that the ascent is slow compared to deep convective systems.

l. 36: Please specify what "This" is referring to.

l. 40: Consider to use "Accordingly" instead of "Thus" to avoid the use of the same wording in two consecutive sentences.

l. 52: Please provide a reference for the statement "persisted for several weeks".

l. 72: Specify here that RASTA is a cloud radar.

l. 87: Is it only the latent heat exchange which is set to zero or are there also other diabatic processes set to zero?

l. 91: Why are you defining three 3-D passive tracers at each grid point and not only one tracer per grid point?

l. 98: According to e.g. Browning et al. (1986), WCBs start to ascend from the planetary boundary layer. In terms of their terminology: Are you really identifying a WCB as it was originally defined or is it convection that is embedded in a slantwise ascending WCB?

l. 106: Please specify that it is 2 October 11:00 UTC.

l. 107: I assume you are meaning "in the eastern half" of the simulation domain. "East of the simulation domain" would actually be outside the domain in Fig. 1.

l. 114: In the region of the cyclonically turning WCB the BT is lower than observed by MSG. In contrast, in NODIA the BTs are similar to the observed values. Do you have any hypothesis why this might be the case?

l. 113-121: It would be very helpful if you labeled some of the key features in Fig. 1 (e.g., cloud head, PV tongue).

l. 132: Please consider to indicate the flight direction (e.g., as an arrow) in Fig. 1a.

l. 141: To my impression the slope also indicates the location of the cold conveyor belt which is located below the cloud shield associated with the WCB. If the authors come to the same conclusion this should be mentioned in the text.

l. 147: Consider to replace "until" with "reaching down to".

l. 160: Can the authors comment on whether this low-level jet is also part of the cold conveyor belt?

l. 162: "close to those measured" is a quite qualitative statement. Could you either show a difference plot of the modeled and observed wind speed or provide a quantitative measure such as RMSE? Also showing a scatter plot of observed vs modeled wind speeds could provide a more quantitative estimate of the differences.

l. 165: Consider to remind the reader that you have selected all ascents with w > 0.3 m s$^{-1}$. Or are you showing air parcels that fulfill the ascent criterion of 150 hPa in 12 h? Please clarify.

l. 171: Also here, a quantitative statement on the differences would be very helpful.

l. 177: Write "profiles" instead of "profile".

l. 180-209: When comparing observations to modeled values at individual grid points, differences might occur due to minor spatial shifts between simulations and observation. To account for these spatial displacements, I suggest to consider the values at several neighboring grid points and to show their variability in Fig. 4. E.g. showing the median value of the grid points together with the interquartile range could be one way to estimate the sampling uncertainty.

l. 215: To my impression there are only two regions of high ascent frequency. One is associated with the bent back warm front and the second region can be found over Greenland. So, what is the reason for splitting the ascent along the bent back warm

front in two regions? Please explain in the text.

l. 223: How did you investigate whether the ascents are produced by the warm front dynamics or by orographic forcing?

l. 233: I assume it is Fig. 6a.

l. 234: I assume it is Fig. 6b.

l. 235: I assume it is Figs. 6a,b

l. 236: I assume it is Fig. 6a.

l. 237: I assume it is Fig. 6b.

l. 239: Correct to Fig. 6a.

l. 240: Correct to Fig. 6c.

l. 299: Why are you referring to the brown circles? As far as I understand correctly, the red stars in Fig. 6a indicate the position of trajectories closest to the time shown in Fig. 9.

l. 307: The rapid segments are not only found in regions of high $\theta_e$, but especially in regions with high $\theta_e$ gradients. This should me mentioned in the discussion.

l. 307 and the following paragraphs: It is not quite clear to me why the focus is on 2 October 2 UTC. The differences between REF and NODIA in terms of upper-tropospheric PV (at 320 K) are considerably larger at 06 UTC. In fact, at 320 K differences in PV at 2 UTC are very difficult to identify. It seems that at 2 UTC the negative PV is mostly located in the mid-troposphere. So, could you comment on the processes leading to the negative PV at 320 K at 06 UTC? Since the differences between REF and NODIA are pronounced at 06 UTC, the negative PV is likely not only a result of isentropic advection.

l. 309: Please provide the coordinates of the rapid segments that are located further

southwestward.

l. 310: Please specify that you are referring to the black dots in Fig. 9b after the statement "... along the bent-back front". In line 311, please clarify that you are referring to the shading in Fig. 9b when discussing the vertical wind speeds.

l. 322: What exactly to you mean by "on the jet stream side".

l. 335: Fig. 10b is a vertical cross section from south to north. So, how is it possible to see the "western edge of the cloudy area"?

l. 354: Can you quantify a bit how much too low?

l. 359: Is this air mass between the warm front and the Greenland plateau really dry? I agree that radar does not detect any precipitation, but I am not convinced that this air mass is dry. Also, it would be interesting to know whether this air mass (especially in the lower troposphere) is the cold conveyor belt of the cyclone.

l. 360: Please explain why the dry air mass is absent. An explanation as in l. 155 would be helpful.

l. 375: Could you explain why the ascents in the WCB outflow are solely due to cloud diabatic processes and not due to frontal dynamics. I think the statement in its current form is very strong and should be reconsidered carefully.

l. 386: To support the statement that especially anticyclonic segments are associated with negative PV: Could you indicate the location of anticyclonic and cyclonic segments in Fig. 9d?

l. 390: Schemm et al. (2013) performed idealized moist and dry simulations of a baroclinic wave. Their results, in particular with respect to the northwestern edge of the ridge are very similar to the results of this study. Please consider to reference their work.

l. 401: The conditional instability is only mentioned here and in the abstract. Please

describe already in the previous Section 5 where exactly the conditional instability can be found. It would be helpful to the reader if the regions of conditional instability were highlighted in the figures or if the latitude longitude coordinates of the unstable regions were provided.

l. 406: This is somewhat related to my previous comment on l. 307. Comparing the evolution of PV at the 320-K isentropic surface in Fig. 8, I have the impression that the negative PV is not simply advected. If this was the case the PV structure should be very similar in REF and NODIA due to conservation of PV in adiabatic flows (Figs. 8c, d). However, REF is characterized by more negative PV in the northwestern corner of the ridge than NODIA. So this clearly points to non-adiabatic processes. My suggestion is that the statement "these structures are then advected by the upper-level anticyclonic flow into the northwestern edge of the ridge" should be extended in the sense that also the non-conservative processes are at least mentioned.

l. 411: A reference for the statement that models "struggle to represent updrafts that do not start in the boundary layer" is needed.

**Figures**

Fig. 1: Please label at least one isobar of the MSLP field in b) and c).

Fig. 5: Please indicate the position of the cyclone center with a marker. This will help the reader to follow the description in Section 4.1. Also, what is the unit of the spatial frequency? Is it simply the total number of air parcels or is it the number of air parcels per area? Please clarify.

Fig. 7: What exactly do mean by "number of rapid segments lies above the average"? Does it mean that it is only shown when more than 50

Fig. 9: "Updrafts and potential vorticity" is a bit confusing since also other parameters are shown. Please consider to remove or replace the first sentence of the caption.

**References**

Browning, K. A., M. E. Hardman, T. W. Harrold, and C. W. Pardoe, 1973: The structure of rainbands within a mid-latitude depression. Quarterly Journal of the Royal Meteorological Society, 99 (420), 215–231, doi:10.1002/qj.49709942002.

Browning, K. A., 1986: Conceptual Models of Precipitation Systems. Wea. Forecasting, 1, 23–41, https://doi.org/10.1175/1520-0434(1986)001<0023:CMOPS>2.0.CO;2.

Eckhardt, S., A. Stohl, H. Wernli, P. James, C. Forster, and N. Spichtinger, 2004: A 15-Year Climatology of Warm Conveyor Belts. J. Climate, 17, 218–237, https://doi.org/10.1175/1520-0442(2004)017<0218:AYCOWC>2.0.CO;2.

Harrold, T. W., 1973: Mechanisms influencing the distribution of precipitation within baroclinic disturbances. Quarterly Journal of the Royal Meteorological Society, 99 (420), 232–251, doi: 10.1002/qj.49709942003

Schemm, S., H. Wernli, and L. Papritz, 2013: Warm Conveyor Belts in Idealized Moist Baroclinic Wave Simulations. J. Atmos. Sci., 70, 627–652, https://doi.org/10.1175/JAS-D-12-0147.1.

---

## Author Comment (AC1) · 11 Jan 2021

We thank the Referee for his/her time and his/her constructive comments. We have complied with most of the proposed changes. In the following, the comments made by the Referee appear in black, while our replies are in blue.

The paper contains a comprehensive case study analysis of convection embedded in a warm conveyor belt and its impact on the upper-level flow. The study combines unique observations taken during the North Atlantic Waveguide and Downstream Impact Experiment and convection-permitting simulations of the case study. The observations are compared to a reference simulation and an experiment in which heat exchanges due to cloud processes are turned off (called NODIA). Generally, the reference simulation agrees with the observations whereas key features are missing in the NODIA experiment, highlighting their diabatic origin. In particular, elongated bands of absolute negative PV are missing in the NODIA simulation. Their impact on the upper-level flow is hence missing in NODIA. These findings support the theory developed in Harvey et al. (2020) and are consistent with those seen in a different cyclone's WCB (Oertel et al. 2020). The case included in this study has been the subject of several recent articles (Maddison et al. 2020, Blanchard et al. 2020), including a recent publications by the authors, and this contribution adds useful new insights to complement the recent research, particularly with the novel observations within the WCB. I thus recommend the article be published subject to minor revisions. I have a couple of broad comments that should be considered before publication and specific and technical comments listed below.

Broad comments:

1) Clarification of online trajectories versus the WCB.

A more careful consideration of how the trajectories shown in the article relate to the WCB ascent would be beneficial. The authors select trajectories in the simulation that ascend 150 hPa in 12 hours (based on the 600 hPa in 48 hour criteria for WCBs used in many other studies). As this is a short time period the trajectories shown don't necessarily correspond to the WCB, as the authors note (section 2.3). As the simulations are run for 36 hours I wonder if there are some trajectories that stay in the domain for longer than 12 hours and could be used to show whether the 12 hour ascents do correspond to part of the WCB or not. Alternatively, successive 12 hour trajectories could be compared in an attempt to "piece together" the WCB flow. This cyclone has been shown to have a WCB (e.g. Maddison et al. 2019) so I would suggest emphasising this (in section 2.3) and terming the ascents "WCB proxy" or something

WCDD
similar. Some properties of the trajectories could then be better explained and would allow for a better placement of the results in the current knowledge. For example, from Figure 7 it appears that the anticyclonic ascents are from the later stages of a WCB ascent (the start at 4km), and the cyclonic ascents from the early part. Also, the characteristic increase and decrease in PV along WCB ascents (e.g. Madonna et al. 2014) is not found here. These should be further explained.

We clarify that the selected ascents "may not all belong to actual WCB trajectories" and refer to Blanchard et al. (2020) for a discussion of the selection criteria. Technically, the scalar tracers used to compute the trajectories are advected during the full 36-h model integration time thus the length of trajectories could be extended. However, the reason for choosing a 12-h window is the domain size, as explained in Section 2.3: "This relatively short time window is chosen to ensure that all relevant trajectories remain in the simulation domain during the 12 h period." (Note that the domain is relatively small compared to earlier WCB studies but still contains 800x800x70 grid points due to the high horizontal resolution.) This is illustrated in Figure 6, where most ascents reaching the red box at the time of observations (11 UTC) are located close to the southern domain boundary at the time of initialization (00 UTC). In particular, the anticyclonic ascents-which "feed" the jet stream core and constitute the WCB outflow-head northward with high velocity at upper levels and their trajectories could be extended by a few hours at most. In contrast, the cyclonic ascents appear to remain longer in the domain but do not contribute to the WCB outflow and jet acceleration thus are not extensively studied here. We clarify that the focus is on the former, especially in Section 5. Finally, the evolution of PV along cyclonic and anticyclonic ascents is further discussed in the text but does not contradict earlier studies (see also response to specific comments).
**2) Verification of the simulations against the observations.**

Throughout the paper the authors compare the reference and NODIA simulations with each other and with the observations. It would be helpful if the authors included some verifications (e.g. RMSE) to clarify and emphasise the comparisons as it is sometimes difficult to see by eye. I would suggest quantifying the simulations' skill in replicating the observed fields in Figures 1, 2 and 3 (comparing points where observations exist). And also comparing the two simulations with each other in Figures 5 and 8. For example, the authors state that the ridge extends further west in the reference simulation so quantifying this somehow (most westward longitude reached for example) would be helpful as it is a bit confusing because of the complicated structure of the ridges. Also the jet stream maxima should be highlighted in the two simulations and discussed more as the title states that the jet stream is accelerated by the convection in the WCB. Several metrics have been added to better compare the simulations and assess them against the observations: the Heidke Skill Score is now computed when comparing Meso-NH BTs with BTs measured by MSG (in addition to already comparing the simulated MSLP with the analysis); quantitative statements are included in the comparison of wind speed between RASTA observations and Meso-NH simulations; finally, the bias and the RMSE are given for the comparison between wind speed, potential temperature and relative humidity measured by the dropsondes and simulated by REF and NODIA.

3) Labelling features of interest.

Several features are referred to in the text that are not always easily recognisable among the highly detailed plots. The authors give latitude or longitude points to guide the reader but this can be quite cumbersome. Adding labels (maybe shapes or simply letters) to the plots for some of the features would help with the comprehension of the
results. The features mentioned in the text that I would suggest labelling include: the high PV tongue, the tropopause fold, the jet cores, the WCB outflow, the bent back front, the low-level jet and the cloud head. Too many labels can of course obscure features and make the plots more complicated but adding one or two labels to some of the figures when latitude/longitude values are needed in the text would be helpful. The suggested features of interest are now labeled in maps and vertical cross-sections in Figs. 1, 2, and 3, while references to geographical coordinates have been omitted when unnecessary.

Specific comments:

L22: PV gradients form a waveguide on zonal flows too (without upper-level ridges or troughs), this should be mentioned here. "Zonal flows" are now mentioned.

L86-88: more information on the other parameterisation schemes in the model should be given here. In particular, would other schemes contain heat exchanges within clouds that would still be active in NODIA? We added "Note that the other parameterizations (radiation, turbulence, shallow convection) also exchange heat in the atmosphere, but in a negligible way compared to cloudy processes."

L128: is this a second MSLP centre (were there two?) or just an eastward movement of the cyclone? Evidence should be provided if it is a second MSLP centre development. The evidence of a second MSLP center is given in Fig. 1b in Blanchard et al. (2020). To clarify, the sentence is now "The abrupt shift is due to the creation of a second MSLP center to the east (see MSLP at 16:00 UTC in Fig. 1b in Blanchard et al. (2020)), which therefore has a diabatic origin."

L206: The fact that the observations are well simulated in REF allows for the attribution of features and their development to diabatic processes. This should be emphasised here. Added
L220: What are the ascents over Greenland associated with? We did not investigate the ascents in detail but their presence in NODIA clearly shows their origin is not diabatic. As explained in the following paragraph "They are likely produced by the combined effect of the warm front dynamics and orographic forcing caused by the Greenland Plateau."

L225-226: I find it surprising that there are almost no ascents in the WCB outflow in NODIA. Is it that the WCB is absent or that the trajectories don't meet the ascent threshold used? It could also be a timing issue in that WCB trajectories may be delayed in NODIA. Would figure 5 look different if a slightly later 12 hour window was chosen? Further explanation should be included here. Indeed they may be trajectories that rise slowly but do not meet the used threshold and thus do not qualify as "ascents" in NODIA. We are actually not surprised as latent heat release associated with cloud diabatic processess—which are switched off in NODIA—are essential to WCB ascents. We added "This absence of trajectories rising by at least 150 hPa in 12 h is consistent with lower cloud tops in NODIA than in REF."

L243: It would be beneficial here if a brief explanation of why/how the anticyclonic trajectories would be expected to impact the upper level flow, via PV modification for example. We added "via injection of low-PV air"

L250-265: can the results be explained here using extratropical cyclone development theory? Does the cyclonic branch of the WCB typically occur later than the anticyclonic? The PV modification along the trajectories is different here than that found in Madonna et al. (2014). There is no increase in PV (as trajectories ascend through heating) and subsequent decrease (as trajectories leave heating). May this have occurred earlier in the ascent? This should be explained here too. The 12-h time window we use for trajectory analysis is too short to compare when the cyclonic and anticyclonic branches of the WCB occur in the Stalactite cyclone, which is beyond the scope of the study. However, for this window, Figure 7 shows that PV actually increases at low levels (cyclonic ascents) and decreases at higher levels (anticyclonic ascents), as exInteractive comment

pected below and above the diabatic heating maximum along slantwise ascents. This was clarified in the text.

L285: Another feature that is clear in Figure 8 is the PV field is smoother in NODIA. This should be mentioned and explained. We do not fully agree: the PV field shows small-scale features in both NODIA and REF (see the cloud head area for example). The main difference between the two simulations lies in the negative PV bands.

L289: mention that the negative PV bands at 06:00 push the ridge cyclonically to the west as well. We added that they "push the ridge to the west" (and omit "cyclonically" to avoid confusion).

L320-326: Why is there no PV dipole for the strong updraft above 6km altitude? Has the PV signature been dissipated by this time? Please explain this here. This may be due to the weaker vertical wind shear but is rather speculative and not discussed in the paper. However, we note that the absence of a PV dipole happens for the strong updraft above 6 km altitude "that does not meet the criteria for rapid segments", which validates the identification of rapid segments based on pressure difference.

L353: Do heat exchanges still occur in other parameterisations? e.g. cloud scheme? In Sect. 2.2., we added "Note that the other parameterizations (radiation, turbulence, shallow convection) also exchange heat in the atmosphere, but in a negligible way compared to cloudy processes." (see response to I. 86–88)

L383: provide some explanation for the rapid ascending trajectories. Mid-level convection is explained below, while low-level rapid segments occurring along cyclonic ascent are not further studied in the paper.

L388: quantify how much further the ridge extends west in REF. The difference looks quite small. Indeed, the difference is small, but still visible. We added "by about 100 km".

L401: is this region of conditional instability shown? Mention if it is or is not. When
we commented on Fig. 9b in Sect. 5, we added "They both lie in a region of vertically homogeneous  $\theta_e$  values, which promotes conditional instability."

Technical comments:

L3: "structures of negative" should be "structures with negative". Changed

L6 (and elsewhere): the authors should explain why the cyclone has been given this name. In the abstract this might not be possible so just saying "a cyclone" here and giving the cyclone its name in the main article may be best. We prefer keeping the name in the abstract, because the cyclone has also been described by other authors, and now explain its origin in the text.

L7-9: I would remove the sentence "The observations reveal..." as the abstract is quite long and this isn't really necessary here. The sentence is crucial to highlight the rare observations and to introduce the double jet stream structure but has been shortened.

L9: change "reproduces well the observed" to "reproduces the observed". Changed

L15: "near the bent back front" in what? The reference simulation? Yes because anticyclonic ascents are absent in the sensitivity experiment as explained in the previous sentence.

L17: remove "and" before "with the negative". Removed

L17: thus appear  $\rightarrow$  the convective cells thus appear. Changed

L17: add "the" before "negative PV bands". Added

L27: reference to Martinez-Alvarado et al. (2018) here. They show Rossby wave amplitude still decreases in more recent NWP model configurations. We now refer to Martinez-Alvarado et al. (2018).

L31-32: change to "WCBs usually flow poleward and upward as coherent...". Changed
L33: band  $\rightarrow$  bands. Changed

L34: clouds  $\rightarrow$  cloud. Changed

L34: During ascents  $\rightarrow$  During WCB ascent. Changed

L35: which representation is  $\rightarrow$  the representation of which is ... . Changed

L39: impact  $\rightarrow$  impacting. Changed

L50: Add why the cyclone is named stalactite. We added "The cyclone was named after the low tropopause—which shape was reminiscent of a stalactite—during its intensification phase."

L106: Add sentence introducing the section and what will be included. We added "An overview of the cloud structures of the Stalactite cyclone and of the associated upper-level ridge is first given."

L116: along  $\rightarrow$  above. Changed

L121: structures  $\rightarrow$  structures present. Changed

L125: Change the sentence "REF reproduces well the ..." to "The track of the Stalactite cyclone is well reproduced in REF". Changed

L128: meridian  $\rightarrow$  meridional. Changed

L145: Highlight where these features are (see broad comment 3). See our response to broad comment 3

L147: 40W until  $z \rightarrow$  40W, reaching z... . Changed

L149: part  $\rightarrow$  part of the domain. Changed

L158: except on  $\rightarrow$  except in. Changed

L159: eastern part where it  $\rightarrow$  eastern part of the domain where it... . Changed
L159: wind speed values  $\rightarrow$  wind speeds. Changed

L162: simulation completes the description  $\rightarrow$  simulation provides a complete description of... . Changed

L166: number ascents  $\rightarrow$  number of ascents. Changed

L177: profile  $\rightarrow$  profiles. Changed

L183-184: might be worth mentioning that the wind speeds in REF still tend to underestimate the observed peak wind speeds. We added "with the exception of slightly underestimated peaks.

L248: remove "a" before higher. altitude  $\rightarrow$  altitudes. Changed

L252: remove "a" before strong. Changed

L255: remove the sentence "some start close to the surface". Removed

L265: swap thereafter with "in the following section". Swapped

L270: track  $\rightarrow$  follow. Changed

L281: "the eastern part of the northwestern edge" is confusing to me. Consider rephrasing. The sentence is now: "In NODIA, the northwestern edge of the ridge and the PV tongue are shifted eastward compared to REF (Fig. 8b)."

L287: merge sentences here: NODIA. But  $\rightarrow$  NODIA, but. Merged

L287: DIA  $\rightarrow$  NODIA. Changed

L288: there  $\rightarrow$  here. Changed

L299: what region is shown in Fig 9 a,b? The red box? We have erroneously referred to the brown circles in Fig. 6a. We now refer to the red stars which indicate the position of trajectories closest to the time shown in Fig. 9 a,b.

L327: remove "Thus". Or join to previous paragraph. The paragraph has been attached
to the previous one.

L359: is this dry air mass the cyclone's dry intrusion? No, that is why we called it "dry air mass", and not dry intrusion to avoid any misunderstanding.

L361: state what the tropopause fold is at the outer boundary of. Changed to "at the edge of the outer part."

L376: explain or motivate why the focus is on the WCB ascents. The sentence has been removed.

L391: Maddison et al. (2020) seems another appropriate reference to add here. Added

L401: "matches with the organised"  $\rightarrow$  "matches the organised". Changed

L406: PV structures are  $\rightarrow$  PV structure in WCB ascent regions are. Changed

References:

be consistent with journal abbreviations. Checked

L423: page and volume numbers missing. Added

Figures:

Fig1: add 'as' before '(a)'. Added

Fig1: What time are the MSLP contours in (b) and (c) shown? At the same time as the BTs. To avoid confusion, the sentence presenting the MSLP contours is now written in second position.

Fig4: mention that the profiles are shown for both observations and simulations in the caption. Information added

WCDD
Fig5: if I understand correctly, the red box is used to select WCB outflow ascents? It is a bit confusing as I initially thought all ascents shown had to have passed through the red box at 11:00? Please clarify this in the text or caption. You understand correctly. In the caption of Fig. 5, it is written that "Spatial frequency [(now) number] of air parcels belonging to the ascents fulfilling the ascent criterion" and "the red box [is] the region where the ascents are selected at 11:00 UTC.".

Fig6: 40 trajectories are plotted, out of how many? Give the number in the text or caption. The number of trajectories is 220 000 for anticyclonic ascents and 250 000 for cyclonic ascents). This information is now written in the caption.

**REFERENCES:**

Oertel, A., Boettcher, M., Joos, H., Sprenger, M., and Wernli, H.: Potential vorticity structure of embedded convection in a warm conveyor belt and its relevance for large-scale dynamics, Weather Clim. Dynam., 1, 127–153, 2020

Maddison, J. W., Gray, S. L., Martínez-Alvarado, O., and Williams, K. D.: Impact of model upgrades on diabatic processes in extratropical cyclones and downstream fore-cast evolution, Quarterly Journal of the Royal Meteorological Society, 146, 1322–1350, 2020.

Blanchard, N., Pantillon, F., Chaboureau, J.-P., and Delanoë, J.: Organization of convective ascents in a warm conveyor belt, Weather Clim. Dynam., 2020

Maddison, J. W., Gray, S. L., Martínez-Alvarado, O., and Williams, K. D.: Upstream Cyclone Influence on the Predictability of Block Onsets over the Euro-Atlantic Region, Monthly Weather Review, 147, 1277–1296, 2019

Madonna, E., Wernli, H., Joos, H. and Martius, O. (2014) Warm conveyor belts in the ERA-Interim dataset (1979–2010). Part I: Climatology and potential vorticity evolution. J. Climate, 27, 3–26.

Martínez-Alvarado, O., Maddison, J. W., Gray, S. L. and Williams, K. D. (2018) At-
mospheric blocking and upper-level Rossby wave forecast skill dependence on model configuration. Quart. J. Roy. Meteor. Soc., 144, 2165–2181.

---

## Author Comment (AC2) · 11 Jan 2021

We thank the Referee for his/her time and his/her constructive comments. We have complied with most of the proposed changes. In the following, the comments made by the Referee appear in black, while our replies are in blue.

Blanchard et al. present a detailed analysis of convection embedded in a WCB and how this affects the upper-tropospheric flow. The study is based on observations taken during the North Atlantic Waveguide and Downstream Impact Experiment and convection-permitting simulations. A reference simulation (REF) generally agrees with the observations and represents key features such as the WCB outflow, a dry region below this outflow and the cloud head associated with the bent-back warm front. A

second simulation is performed with latent heating exchanges due to cloud processes being turned off (NODIA). A comparison of the two simulations reveals that elongated bands of negative PV are missing the the NODIA simulation pointing to their diabatic origin. Indeed, the analysis of trajectories and vertical cross section through the WCB suggests that mid-level convection embedded in the WCB is responsible for generating the bands of negative PV in a vertically sheared environment. This is in line with recent studies by Harvey et al. (2020) and Oertel et al. (2020). The study is well written, the figures are mostly clear and the methods are sound. As the paper confirms recent research using novel observations and a slightly different approach (simulations with latent heat release switched on/off), I recommend the article to be published in WCDD after the following comments have been addressed.

**Broad comments**

1) The REF and NODIA simulations are compared qualitatively throughout the paper. To my impression it would be helpful if the authors provided quantitative estimates of the differences between the simulations since it is sometimes difficult to spot the differences by eye. As an alternative, difference plots would help the reader to fully appreciate the differences (e.g, Fig. 3, 8) which are discussed in the text.
As also suggested by Referee 1, the Heidke Skill Score is now computed when comparing brightness temperatures simulated by Meso-NH and measured by MSG, quantitative statements are included in the comparison of wind speed between RASTA observations and Meso-NH simulations, while the bias and the root-mean square error are given for the comparison between wind speed, potential temperature and relative humidity measured by the dropsondes and simulated by REF and NODIA.

2) The individual subsections are quite often introduced by describing what is shown in the figures. These descriptions are not necessary since they are also provided in the figure captions. Instead, it would be helpful if the authors described the purpose of each subsection in one to two sentences. This would help to guide the reader through the manuscript.

As suggested, the subsection headers have been rephrased to introduce their topic rather than the figures they describe.

**Minor comments**

l. 2: Please clarify that "their" is referring to WCBs and not to "ridges". Changed to "the representation of WCBs"

l. 9: Since the "mesoscale structures" are mentioned here for the first time. Please specify what the "mesoscale structures" are. Are these the tropopause fold and the jet stream core? We removed "mesoscale" as we refer to the "fine-scale observations of cloud and wind structures acquired with airborne Doppler radar and dropsonde"

l. 22: Also PV gradients along zonal flows form a waveguide. Please include this as well. Included

l. 32: I'd suggest to also cite at least one of the early studies, e.g., by Browning et al. (1973) and Harrold (1973). The study of Harrold (1973) is now cited.

l. 32: Other studies state that WCBs are characterized by "rapid ascent" (e.g., Eckardt et al. 2004). Compared to deep convection the WCB ascent may be considered as "slow". Perhaps specify that the ascent is slow compared to deep convective systems. We removed "slowly"

l. 36: Please specify what "This" is referring to. Changed to "This source"

l. 40: Consider to use "Accordingly" instead of "Thus" to avoid the use of the same wording in two consecutive sentences. Changed

l. 52: Please provide a reference for the statement "persisted for several weeks". The reference to Schäfler et al. (2018) has been added.

l. 72: Specify here that RASTA is a cloud radar. Added

l. 87: Is it only the latent heat exchange which is set to zero or are there also other

diabatic processes set to zero?  We added "Note that the other parameterizations (radiation, turbulence, shallow convection) also exchange heat in the atmosphere, but in a negligible way compared to cloudy processes."

l. 91: Why are you defining three 3-D passive tracers at each grid point and not only one tracer per grid point? We added "Three scalar tracers per grid point allow to follow the three dimensional position of each air parcel."

l. 98: According to e.g. Browning et al. (1986), WCBs start to ascend from the planetary boundary layer. In terms of their terminology: Are you really identifying a WCB as it was originally defined or is it convection that is embedded in a slantwise ascending WCB? We do not claim to formally identify a WCB. As stated, "Selected ascents thus do not perform a full ascent from the boundary layer to the upper troposphere". We clarify that they "may not all belong to actual WCB trajectories" and refer to Blanchard et al. (2020) for a discussion.

l. 106: Please specify that it is 2 October 11:00 UTC. Added

l. 107: I assume you are meaning "in the eastern half" of the simulation domain. "East of the simulation domain" would actually be outside the domain in Fig. 1. Corrected

l. 114: In the region of the cyclonically turning WCB the BT is lower than observed by MSG. In contrast, in NODIA the BTs are similar to the observed values. Do you have any hypothesis why this might be the case? Thanks for pointing this. The underestimation of the BTs was an artifact due to the cloud properties that were used to compute the BTs. It has been corrected.

l. 113-121: It would be very helpful if you labeled some of the key features in Fig. 1 (e.g., cloud head, PV tongue). Added

l. 132: Please consider to indicate the flight direction (e.g., as an arrow) in Fig. 1a. Added

l. 141: To my impression the slope also indicates the location of the cold conveyor belt

which is located below the cloud shield associated with the WCB. If the authors come to the same conclusion this should be mentioned in the text. We agree and mention this in the text.

l. 147: Consider to replace "until" with "reaching down to". Changed to "reaching"

l. 160: Can the authors comment on whether this low-level jet is also part of the cold conveyor belt? We commented that "The low-level jet likely corresponds to the cold conveyor belt with possible orographic influence."

l. 162: "close to those measured" is a quite qualitative statement. Could you either show a difference plot of the modeled and observed wind speed or provide a quantitative measure such as RMSE? Also showing a scatter plot of observed vs modeled wind speeds could provide a more quantitative estimate of the differences. We prefer to keep focus on the impact of the cloud diabatic impact. However, we added "with a bias of $0.5\,\mathrm{m}\,s^{-1}$ and the root-mean square error of $3.3\,\mathrm{m}\,s^{-1}$" to provide the reader with a quantitative statement.

l. 165: Consider to remind the reader that you have selected all ascents with $w > 0.3$ m s$^{-1}$. Or are you showing air parcels that fulfill the ascent criterion of 150 hPa in 12h? Please clarify. We added "(that fulfill the ascent criterion of 150 hPa in 12 h)".

l. 171: Also here, a quantitative statement on the differences would be very helpful. We added "The maximum value is equal to $38\,\mathrm{m}\,s^{-1}$, a value lower than the maximum of $42\,\mathrm{m}\,s^{-1}$ obtained for REF."

l. 177: Write "profiles" instead of "profile". Changed

l. 180-209: When comparing observations to modeled values at individual grid points, differences might occur due to minor spatial shifts between simulations and observation. To account for these spatial displacements, I suggest to consider the values at several neighboring grid points and to show their variability in Fig. 4. E.g. showing the median value of the grid points together with the interquartile range could be one way to

Interactive
comment

estimate the sampling uncertainty. As one may expect, the simulated fields are rather smooth compared to observations from radiosondes (see curves on Fig. 4). They show zonal gradients (see Fig. 3) but these precisely allow to assess the horizontal extent of the simulated features. For these reasons, and for the sake of visibility, we prefer to show the simulated values at the nearest grid point only. This further allows us to calculate the bias and root-mean square error between the dropsonde measurements and the simulated values for wind speed, potential temperature and relative humidity.

l. 215: To my impression there are only two regions of high ascent frequency. One is associated with the bent back warm front and the second region can be found over Greenland. So, what is the reason for splitting the ascent along the bent back warm front in two regions? Please explain in the text. We do not share your impression, because the area north of the cyclone does not overlap with the bent-back front. To illustrate this point, we have added "(as shown in Sec. 3.2)" after "It corresponds to the WCB outflow region overflown by the aircraft".

l. 223: How did you investigate whether the ascents are produced by the warm front dynamics or by orographic forcing? We did not investigate their origin in detail but their presence in NODIA clearly shows it is not diabatic. The sentence has been rephrased and is now "They are likely produced by the combined effect of the warm front dynamics and orographic forcing caused by the Greenland Plateau."

l. 233: I assume it is Fig. 6a. Corrected

l. 234: I assume it is Fig. 6b. Corrected

l. 235: I assume it is Figs. 6a,b. Corrected

l. 236: I assume it is Fig. 6a. Corrected

l. 237: I assume it is Fig. 6b. Corrected

l. 239: Correct to Fig. 6a. Corrected
l. 240: Correct to Fig. 6c. Corrected

l. 299: Why are you referring to the brown circles? As far as I understand correctly, the red stars in Fig. 6a indicate the position of trajectories closest to the time shown in Fig. 9. Changed to "red stars"

l. 307: The rapid segments are not only found in regions of high $\theta_e$, but especially in regions with high $\theta_e$ gradients. This should be mentioned in the discussion. Added

l. 307 and the following paragraphs: It is not quite clear to me why the focus is on 2 October 2 UTC. The differences between REF and NODIA in terms of upper-tropospheric PV (at 320 K) are considerably larger at 06 UTC. In fact, at 320 K differences in PV at 2 UTC are very difficult to identify. It seems that at 2 UTC the negative PV is mostly located in the mid-troposphere. So, could you comment on the processes leading to the negative PV at 320 K at 06 UTC? Since the differences between REF and NODIA are pronounced at 06 UTC, the negative PV is likely not only a result of isentropic advection. We agree that differences on the 320-K isentropic level are larger at 06 UTC. However, the negative PV bands have already formed and convection has weakened at that time. We clarified the focus on the early hours at the beginning of the Section: "The origin of the negative PV bands is now investigated in the region where both the anticyclonic ascents start (red stars in Fig. 6a) and the elongated negative PV bands found in the WCB outflow region appear to form (box in Fig. 8e). Furthermore, time evolutions have shown that anticyclonic rapid segments are most numerous during the early simulation hours (see black boxplots in Fig. 7a)."

l. 309: Please provide the coordinates of the rapid segments that are located further southwestward. We added "(around $56°$ N and $30–31°$ W)"

l. 310: Please specify that you are referring to the black dots in Fig. 9b after the statement "... along the bent-back front". In line 311, please clarify that you are referring to the shading in Fig. 9b when discussing the vertical wind speeds. Changed following your suggestions.

l. 322: What exactly to you mean by "on the jet stream side". Changed to "facing the jet stream core"

l. 335: Fig. 10b is a vertical cross section from south to north. So, how is it possible to see the "western edge of the cloudy area"? Changed to "southern edge of the cloudy area"

l. 354: Can you quantify a bit how much too low? The sentence is now "...whereas the cloud tops are generally 1 km too low in NODIA."

l. 359: Is this air mass between the warm front and the Greenland plateau really dry? I agree that radar does not detect any precipitation, but I am not convinced that this airmass is dry. Also, it would be interesting to know whether this air mass (especially in the lower troposphere) is the cold conveyor belt of the cyclone. We agree that the cloud radar observation can only infer the absence of clouds. However, dropsondes show relative humidity as low as 20%. The air mass is therefore quite dry. The sentence is now "These observations combined with dropsonde measurements ..."

l. 360: Please explain why the dry air mass is absent. An explanation as in l. 155 would be helpful. We added "(cutting off the diabatic cooling reduces evaporation of frozen hydrometeors under the warm front)".

l. 375: Could you explain why the ascents in the WCB outflow are solely due to cloud diabatic processes and not due to frontal dynamics. I think the statement in its current form is very strong and should be reconsidered carefully. Changed to "ascents in the WCB outflow do not occur in the absence of cloud diabatic processes."

l. 386: To support the statement that especially anticyclonic segments are associated with negative PV: Could you indicate the location of anticyclonic and cyclonic segments in Fig. 9d? At 02:00 UTC, rapid anticyclonic segments are located at around 4 km altitude, while rapid cyclonic segments are almost absent (Fig. 7). The former correspond well to the updrafts and to the negative PV values in Fig. 9, while the (few) latter are

located further westward and not shown for the sake of clarity.

l. 390: Schemm et al. (2013) performed idealized moist and dry simulations of a baroclinic wave. Their results, in particular with respect to the northwestern edge of the ridge are very similar to the results of this study. Please consider to reference their work. Added

l. 401: The conditional instability is only mentioned here and in the abstract. Please describe already in the previous Section 5 where exactly the conditional instability can be found. It would be helpful to the reader if the regions of conditional instability were highlighted in the figures or if the latitude longitude coordinates of the unstable regions were provided. In the comment on Fig. 9b in Sect. 5, we added "They both lie in a region of vertically homogeneous $\theta_e$ values, which promotes conditional instability."

l. 406: This is somewhat related to my previous comment on l. 307. Comparing the evolution of PV at the 320-K isentropic surface in Fig. 8, I have the impression that the negative PV is not simply advected. If this was the case the PV structure should be very similar in REF and NODIA due to conservation of PV in adiabatic flows (Figs. 8c, d). However, REF is characterized by more negative PV in the northwestern corner of the ridge than NODIA. So this clearly points to non-adiabatic processes. My suggestion is that the statement "these structures are then advected by the upper-level anticyclonic flow into the northwestern edge of the ridge" should be extended in the sense that also the non-conservative processes are at least mentioned. We agree. As tracking the negative PV structures would require different tools than the Lagrangian trajectories and lies beyond the scope of the study, we simply state that the strcututres are "transported by the anticyclonic flow" into the northwestern edge of the ridge.

l. 411: A reference for the statement that models "struggle to represent updrafts that do not start in the boundary layer" is needed. We now refer to the study of McTaggart-Cowan et al. (2020).

**Figures**

[Figure]

Fig. 1: Please label at least one isobar of the MSLP field in b) and c). Added

Fig. 5: Please indicate the position of the cyclone center with a marker. This will help the reader to follow the description in Section 4.1. Also, what is the unit of the spatial frequency? Is it simply the total number of air parcels or is it the number of air parcels per area? Please clarify. The position of the cyclone center is now shown with "L" and "spatial frequency" is now "number of air parcels".

Fig. 7: What exactly do mean by "number of rapid segments lies above the average"? Does it mean that it is only shown when more than 50. Change to "above their time average".

Fig. 9: "Updrafts and potential vorticity" is a bit confusing since also other parameters are shown. Please consider to remove or replace the first sentence of the caption. Changed to "Results"

**References**

Browning, K. A., M. E. Hardman, T. W. Harrold, and C. W. Pardoe, 1973: The structure of rainbands within a mid-latitude depression. Quarterly Journal of the Royal Meteorological Society, 99 (420), 215–231, doi:10.1002/qj.49709942002.

Browning, K. A., 1986: Conceptual Models of Precipitation Systems. Wea. Forecasting, 1, 23–41, https://doi.org/10.1175/1520-0434(1986)001<0023:CMOPS>2.0.CO;2.

Eckhardt, S., A. Stohl, H. Wernli, P. James, C. Forster, and N. Spichtinger, 2004: A 15-Year Climatology of Warm Conveyor Belts. J. Climate, 17, 218–237,https://doi.org/10.1175/1520-0442(2004)017<0218:AYCOWC>2.0.CO;2.

Harrold, T. W., 1973: Mechanisms influencing the distribution of precipitation within baroclinic disturbances. Quarterly Journal of the Royal Meteorological Society, 99(420), 232–251, doi: 10.1002/qj.49709942003

McTaggart-Cowan, R., Vaillancourt, P. A., Separovic, L., Corvec, S., and Zadra, A.,

2020: A Convection Parameterization for Low-CAPE Environments, Mon. Weather Rev., in press, https://doi.org/10.1175/MWR-D-20-0020.1

Schäfler, A., Craig, G., Wernli, H., Arbogast, P., Doyle, J. D., McTaggart-Cowan, R., Methven, J., Rivière, G., Ament, F., Boettcher, M.,Bramberger, M., Cazenave, Q., Cotton, R., Crewell, S., Delanoë, J., Dörnbrack, A., Ehrlich, A., Ewald, F., Fix, A., Grams, C. M., Gray,S. L., Grob, H., Groß, S., Hagen, M., Harvey, B., Hirsch, L., Jacob, M., Kölling, T., Konow, H., Lemmerz, C., Lux, O., Magnusson, L.,Mayer, B., Mech, M., Moore, R., Pelon, J., Quinting, J., Rahm, S., Rapp, M., Rautenhaus, M., Reitebuch, O., Reynolds, C. A., Sodemann,465H., Spengler, T., Vaughan, G., Wendisch, M., Wirth, M., Witschas, B., Wolf, K., and Zinner, T.: The North Atlantic Waveguide andDownstream Impact Experiment, Bull. Amer. Meteo. Soc., 99, 1607–1637, https://doi.org/10.1175/BAMS-D-17-0003.1, 2018

Schemm, S., H. Wernli, and L. Papritz, 2013: Warm Conveyor Belts in Idealized Moist Baroclinic Wave Simulations. J. Atmos. Sci., 70, 627–652, https://doi.org/10.1175/JAS-D-12-0147.1.